# Botulinum toxin intoxication requires retrograde transport and membrane translocation at the ER in RenVM neurons

**Jeremy C Yeo[1], Felicia P Tay[1], Rebecca Bennion[2], Omar Loss[3], Jacquie Maignel[4], Laurent Pons[4], Keith Foster[3], Matthew Beard[3], Frederic Bard[1,2]\***

[1]Institute of Molecular and Cell Biology, Singapore, Singapore; [2]Centre de Recherche en Cancérologie de Marseille, Aix Marseille Université, Inserm, CNRS, Institut Paoli-Calmettes, Equipe Leader Fondation ARC 2021, Marseille, France; [3]Ipsen Bioinnovation, London, United Kingdom; [4]Ipsen Innovation, Les Ulis, France

**\*For correspondence:**
frederic.bard@inserm.fr

**Abstract** Botulinum neurotoxin A (BoNT/A) is a highly potent proteolytic toxin specific for neurons with numerous clinical and cosmetic uses. After uptake at the synapse, the protein is proposed to translocate from synaptic vesicles to the cytosol through a self-formed channel. Surprisingly, we found that after intoxication proteolysis of a fluorescent reporter occurs in the neuron soma first and then centrifugally in neurites. To investigate the molecular mechanisms at play, we use a genome-wide siRNA screen in genetically engineered neurons and identify over three hundred genes. An organelle-specific split-mNG complementation indicates BoNT/A traffic from the synapse to the soma-localized Golgi in a retromer-dependent fashion. The toxin then moves to the ER and appears to require the Sec61 complex for retro-translocation to the cytosol. Our study identifies genes and trafficking processes hijacked by the toxin, revealing a new pathway mediating BoNT/A cellular toxicity.

## eLife assessment

In this **valuable** manuscript, Yeo et al. describe new methods for assessing the intracellular itinerary of Botulinum neurotoxin A (BoNT/A), a potent toxin used in clinical and cosmetic applications. The current manuscript challenges previously held views on how the catalytic portion of the toxin makes its way from the endocytic compartment to the cytosol, to meet its substrates. The approach taken is deemed innovative and the experiments are carefully performed, presenting **solid** evidence for some of the drawn conclusion; however, the conclusions one may draw from the experimental results are somewhat limited, as it is possible that the scope of their findings could be restricted to the specific neuron model and molecular tools that were used. This paper could be of interest to both cell biologists and physicians.

## Introduction

BoNTs are the most potent toxins known, with a lethal dose of less than a microgram; paradoxically they are also one of the most widely used drugs today for neuromuscular conditions and various aesthetic procedures. BoNTs block overactive muscles involved in neuromuscular disorders or wrinkles by preventing nerve stimulation. The high potency of BoNTs allows for precise localized treatments through the injection of small quantities of toxins. Their long-lasting effects make them attractive treatments despite the required injection. These remarkable properties of BoNTs are derived from their unique intoxication biology. Yet, key features of this biology remain poorly understood.

BoNTs target synapses at the neuro-muscular junction, where they cleave specific cytosolic proteins involved in the release of the acetylcholine neurotransmitter in neurons (*Dong et al., 2019*; *Montecucco and Schiavo, 1994*). As BoNTs are proteins of ~150 kDa, they cannot diffuse through cellular membranes. The key biological features of BoNTs are: (1) high binding specificity for neurons, specifically neuronal synapses, (2) ability to reach the cytosol of neurons, and (3) high specificity for a molecular target once in the cytosol. Of these three features, the second is probably the least understood.

BoNTs are naturally expressed by the gram-positive bacterium *Clostridium botulinum* (*C. botulinum*) as a single ~150 kDa polypeptide chain, then post-translationally cleaved to produce a 50 kDa N-terminal catalytic light (L) chain, which is a zinc metalloprotease and a 100 kDa C-terminal heavy (H) chain. The L and H chains are linked via a disulfide bond and other noncovalent interactions to form the active toxin. *C. botulinum* produces seven serotypes of BoNTs, A to G, and while they all interfere with neurotransmission, they have different cell surface receptors and, interestingly, different kinetics of action across subtypes (*Pellett et al., 2015b*; *Rasetti-Escargueil and Popoff, 2020*).

BoNT/A is the most clinically-used serotype, it binds to the synaptic membrane of neurons with high specificity, with synergistic binding to the synaptic vesicle protein 2 (SV2) and the trisialoganglioside GT1b (*Dong et al., 2006*; *Yowler and Schengrund, 2004*). After internalization and translocation to the cytosol, the light chain of BoNT/A binds and cleaves the SNARE (Soluble N-ethylmaleimide-sensitive factor Attachment protein REceptor) SNAP25 between the residues Q197 and R198. As SNAP25 is essential for the release of synaptic vesicles, its inactivation results in the inability of neurons to release acetylcholine, the key neurotransmitter at neuromuscular junctions (*Dong et al., 2019*; *Montecucco and Schiavo, 1994*). As the toxin proteolytic activity is highly specific, the neuron remains viable and able to recover once the toxin has been cleared (*Peng et al., 2013*).

The current model for intoxication postulates cell surface binding and internalization in endosomes, then, due to changes in pH, there are conformational changes in the heavy chain that leads to the formation of a channel which ushers the L chain into the cytosol of the axonal bouton (*Azarnia Tehran et al., 2017*; *Pirazzini et al., 2014*). The heavy chain is proposed to contain an N-terminal translocation domain (HN) in addition to the C-terminal receptor-binding domain (HC). Once or while the L chain is translocated, it is released from the H chain by disulfide bond reduction, mediated by the thioredoxin reductase 1 (TXNRD1) (*Dong et al., 2019*; *Pirazzini et al., 2013*; *Rossetto et al., 2021*). After refolding of the L chain, it binds and cleaves SNAP25 at a specific residue.

Parts of this scenario have been challenged by studies showing that there are no distinct structural changes to the H and L domains upon acidification and by the difficulty in observing in vitro a self-translocation process (*Araye et al., 2016*; *Galloux et al., 2008*). Studies have also shown that BoNT/A is trafficked beyond the axon boutons in non-acidic and non-recycling vesicles (*Antonucci et al., 2008*; *Harper et al., 2016*; *Restani et al., 2012*). In addition, the current model fails to explain the delayed onset of action, with hours to days between the time of injection and muscle paralysis (*Ledda et al., 2022*). Arguably, the main problem of the model is its failure to propose a thermodynamically consistent explanation for the directional translocation of a polypeptidic chain across a biological membrane. Other known instances of polypeptide membrane translocation such as the co-translational translocation into the ER indicate that it is an unfavorable process, which requires cellular energy (*Alder and Theg, 2003*).

Other similar toxins, like the Cholera toxin, Ricin and Pseudomonas exotoxin A, follow a complex intracellular trafficking route, first from endosomes to the Golgi apparatus, then to the endoplasmic reticulum where they translocate across the membrane. The translocation event itself relies on the host translocon machinery or other ER endogenous complexes (*Moreau et al., 2011*; *Nowakowska-Gołacka et al., 2019*; *Zhang et al., 2013*).

In this study, we present a novel assay based on genetically modified human progenitors able to generate functional neurons (*Donato et al., 2007*; *Song et al., 2019*). The reporter assesses BoNT/A proteolytic activity in live cells and is amenable to a high throughput screening. Complete cleavage of the reporter requires nearly 72 hr of exposure. Surprisingly, we find that BoNT/A proteolytic activity is first detected in the soma of neurons, 24–48 hr before reaching the end of neurites. To investigate the underlying reasons, we performed a genome-wide RNAi survey of the host factors required for BoNT/A intoxication. Our results reveal a high number of genes linked to membrane trafficking, suggesting that BoNT/A follows a complex intracellular route. We use another set of reporters, based on mNG fluorescence reconstitution to illustrate that retro-axonal traffic is required for BoNT/A to

reach the Golgi, then the ER, where it translocates to the cytosol using the Sec61 translocon. These findings help explain the delayed effect of the toxin and could pave the way to improved therapeutics.

## Results

### ReD SNAPR: Neuronal cells expressing a BoNT/A reporter derived from SNAP25

To establish a high-throughput assay for BoNT/A activity, we selected an abundant and consistent source of neuronal cells, the ReNcell VM, a v-myc-transformed human neuronal stem cell line. This cell line has the ability to differentiate into neurons in about two weeks after withdrawal of EGF and bFGF from the culture medium. Over this period, neurites are formed and the neuronal markers, beta3-tubulin, and MAP2 increase significantly in differentiated cells (*Figure 1—figure supplement 1A*). In parallel, the low levels of the oligodendrocyte marker CNPase expressed in the stem cells are further diminished.

To detect BoNT/A activity, we generated a chimeric reporter protein composed of SNAP25 flanked by the red fluorescent protein called tagRFP (tRFP) and the green fluorescent protein tagGFP (tGFP) at its N-terminus and C-terminus, respectively (*Shaner et al., 2008*). The construct was named SNAPR (<u>SNAP</u>25 tagged with <u>R</u>FP and GFP). Using lentiviral transduction, we generated ReNcell VM cells stably expressing SNAPR (*Figure 1A*). We coined this cell line Red-SNAPR for <u>Re</u>Ncell-<u>d</u>erived, expressing <u>SNAPR</u>. After BoNT/A cleaves the SNAP25 moiety at Q197R, the C-terminal tGFP-containing moiety is rapidly degraded while the rest of the construct is preserved (*Figure 1A*).

After incubation with 100 nM BoNT/A for 48 hr, tGFP fluorescence was noticeably diminished compared to tRFP, whose signal remains stable. By western blot, a ~75 kDa SNAPR construct was detected by both tRFP and tGFP antibodies, and cleaved to a 50 kDa product only detected by the tRFP antibody, while the expected ~25 kDa tGFP fragment was undetectable (*Figure 1B*). This stable cell line has unaltered differentiation potential (*Figure 1—figure supplement 1B*).

To test whether BoNT/A intoxication was dependent on neuronal activity, we used the neurotrophic factors GDNF and BDNF to enhance neuronal differentiation. We further supplemented the medium with high salt (KCl and CaCl) for neuronal stimulation (*Harper et al., 2011*; *Pellett et al., 2015a*). This differentiation and stimulation media (ReDS media) resulted in an improvement of sensitivity by an order of magnitude in the imaging assay (*Figure 1D and E*). This increased sensitivity was confirmed by the western blot in which the $EC_{50}$ of BoNT/A intoxication improved from 43 nM to 6 nM (*Figure 1F*). This strongly suggests that toxin binding and uptake occur at the level of active synapses, as is the case in vivo.

### BoNT/A activity is first detected in the soma of neurons

During the initial experiments, we noticed that optimal degradation of the reporter required 48 hr after adding the toxin. This delay was surprising considering the model of rapid translocation after internalization. As the reporter allows live detection of BoNT/A proteolytic activity, we were curious to observe the distribution of BoNT/A activity within the neurons over time. We thus imaged cells at 24, 48, and 72 hr after intoxication. To facilitate the imaging of individual neurites, a co-culture of Red-SNAPR to ReNcell VM cells (1:4 ratio) was implemented (*Figure 2A*). Surprisingly, at 24 hr. there was no loss of GFP signal in the terminal part of neurites but visible degradation at the neurite hillock. At 36 hr, tGFP degradation progressed towards the axon terminals and most of the tGFP signal was eliminated by 48 hr. The pattern suggests a slow distribution of BoNT/A from the cell body to the terminus of axons (i.e. ~16 µm/hr) (*Figure 2B*).

### BoNT/A protein is also detected first in neuronal soma

SNAPR reports on BoNT activity and it is conceivable that BoNT/A might not be active immediately after translocation, thus potentially affecting the spatiotemporal pattern of proteolytic activity. To directly detect BoNT protein in the cytosol, we generated a ReNcell VM cell line expressing an HA-tagged split monomeric NeonGreen (mNG) protein targeted to the cytosol (Cyt-mNG$_{1-10}$) (*Feng et al., 2017*). We verified that Cyt-mNG$_{1-10}$ was expressed using the HA tag, the expression was homogeneously distributed in differentiated neurons and we observed no mNG signal (*Figure 2C*). We also

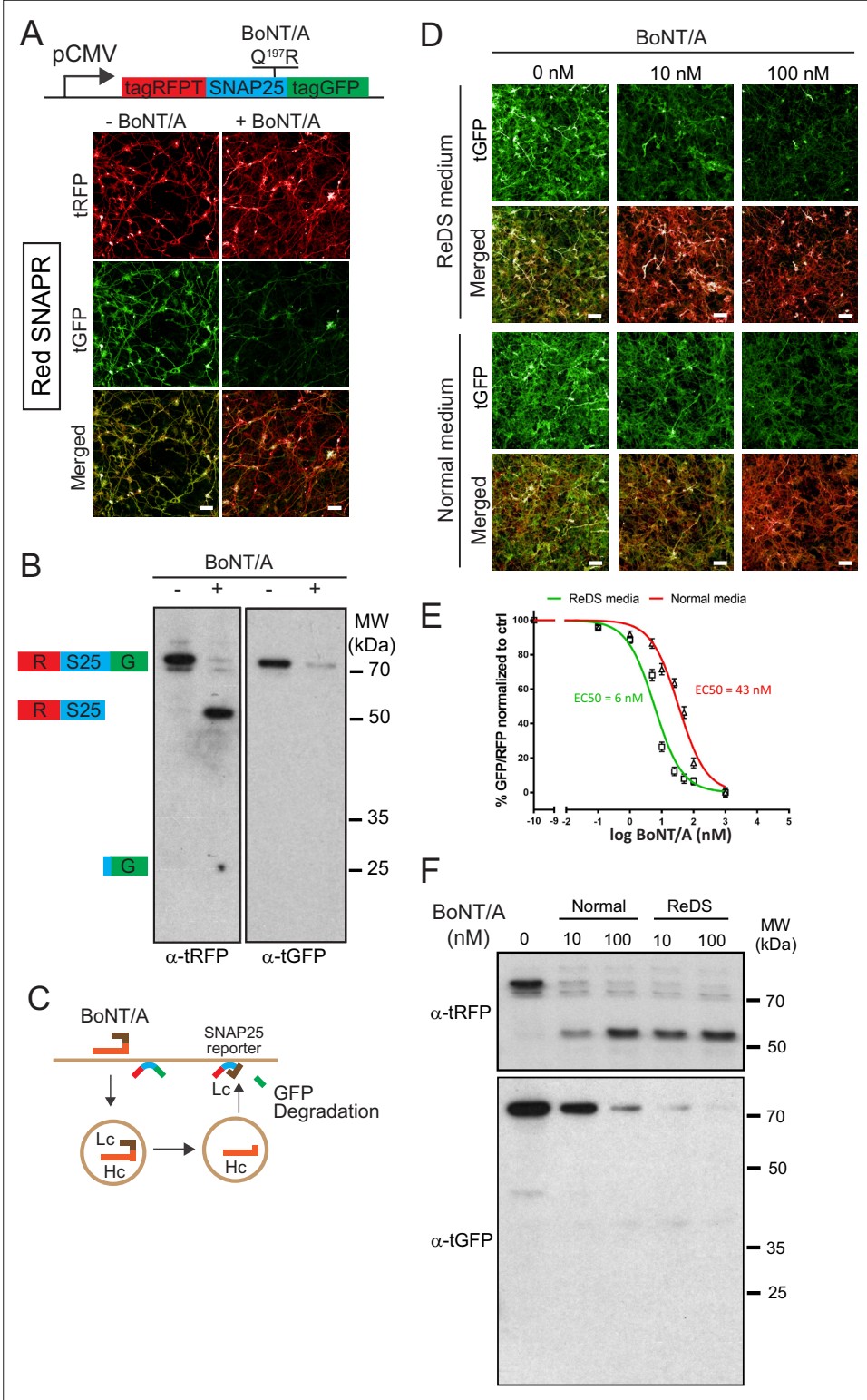

**Figure 1.** Differentiated Botulinum neurotoxin A (BoNT/A) reporter cell line, Red SNAPR, is highly sensitive to BoNT/A intoxication. (**A**) Schematic diagram of the BoNT/A reporter construct, SNAPR. Representative images of ReNcell VM cell line, Red SNAPR, stably expressing SNAPR and incubated with 100 nM BoNT/A for 48 hr. (**B**) Western blot of cell lysates from (**A**) probed with tRFP and tGFP antibodies. (**C**) Schematic diagram of the BoNT/A intoxication assay. (**D**) Red SNAPR differentiated in normal or ReDS medium, then incubated with 0, 10, and 100 nM BoNT/A for 48 hr. (**E**) Quantification of EC$_{50}$ dose response of BoNT/A in ReD SNAPR cells incubated with

*Figure 1 continued on next page*

*Figure 1 continued*

0–100 nM BoNT/A in normal and ReDS medium using GFP/RFP ratio readout. Mean + SEM with n = 3 experiments with at least 200 cells from each experiment. (**F**) Western blot of cell lysates from (**D**) probed with tRFP and tGFP antibodies. Scale bars 50 μm.

The online version of this article includes the following source data and figure supplement(s) for figure 1:

**Source data 1.** PDF containing original images for the western blot in *Figure 1B* (anti-tRFP and anti-tGFP).

**Source data 2.** PDF containing original images of the western blot in *Figure 1B* (anti-tRFP and anti-tGFP) with highlighted bands and sample labels.

**Source data 3.** PDF containing original images for the western blot in *Figure 1F* (anti-tRFP and anti-tGFP).

**Source data 4.** PDF containing original images of the western blot in *Figure 1F* (anti-tRFP and anti-tGFP) with highlighted bands and sample labels.

**Figure supplement 1.** Differentiation and siRNA depletion dynamics in ReNcell VM.

generated and produced the complementary $mNG_{11}$ fused to BoNT/A (BoNT/A-$mNG_{11}$) with three beta-strands of mNG in tandem (*Figure 2C*).

We next imaged the neurons after different intoxication times (*Figure 2D*). After 12 hr, the first signal was observed in the cell body of neurons, in concordance with the pattern of BoNT activity. The fluorescence appeared in specks first, then converted to a more diffuse pattern. At 24 hr and 36 hr, the mNG pattern became more diffuse and started to spread in the neurites. The initial pattern might reflect the reconstitution of mNG proteins at the site of extrusion from membranes. By 48 hr, the mNG signal had filled up the neurites, indicating that BoNT/A had diffused throughout the cell (*Figure 2D*). Quantification of the mNG signal confirmed an initial accumulation in the soma followed by an increase in neurites (*Figure 2D*). The late (>24 hr) and gradual accumulation in neurites was further confirmed by quantification of intensity correlation between BoNT/A LC-mNG and α-Tuj1, a neuronal marker enriched in neurites (*Figure 2D*).

Altogether, the data indicates that BoNT/A translocates into the cytosol at the level of the soma. Based on the dependency of the toxin on neuronal activity and current knowledge of BoNT/A receptors, it is likely that BoNT/A is internalized at the tip of neurites, where synapses form. Thus, the data would suggest that BoNT/A requires retrograde trafficking before it can translocate.

## A genome-wide RNAi screen reveals numerous positive and negative regulators of BoNT intoxication

To elucidate the molecular mechanisms required for BoNT/A intoxication and understand the surprising pattern of appearance, we decided to systematically survey the genes required for BoNT/A translocation.

We first optimized the liposome-mediated delivery of siRNA in the Red-SNAPR cells (*Figure 1—figure supplement 1C*). As a positive control, we targeted TXNRD1 as a factor known to be required. Cells differentiated for 2 weeks were transfected with TXNRD1 siRNA for 3–5 days and analyzed by image analysis. Undifferentiated and differentiated ReNcell VM displayed ~70% knockdown compared to non-targeting control as measured by image analysis of TXNRD1 staining. We also targeted SNAPR using an siRNA targeting SNAP25. This approach achieved an 80% reduction in signal (*Figure 1—figure supplement 1D*).

As neurons cannot be passaged after differentiation, a forward siRNA transfection pipeline was developed (*Figure 3A*). The genome-wide screen was carried out in duplicate using pools of 4 siRNAs per gene and targeting 21,121 human genes in total. Differentiated Red-SNAPR cells on laminin-coated 384-well imaging plates were incubated with siRNA complexes for 3 days and intoxicated with BoNT/A for 2 days before imaging. The positive control siTXNRD1 rescued the tGFP signal reproducibly (*Figure 3B*).

We analyzed the two replicates of the whole genome-wide screen using the ScreenSifter software (*Kumar et al., 2013*; *Figure 3—figure supplement 1A*). The data was converted to average plate Z-score, revealing low plate-to-plate variations (*Figure 3C*). The controls such as the BoNT/A--treated, siNT3 (NT3+), non BoNT/A-treated siNT3 (NT3-) and BoNT/A-treated, siTXNRD1 (TXNRD +in figure) were tightly grouped (*Figure 3D*). The assay had a robust Z-factor of 0.81. The two replicates were correlated with an R-value of 0.74. Genome-wide plots for individual replicates are shown in

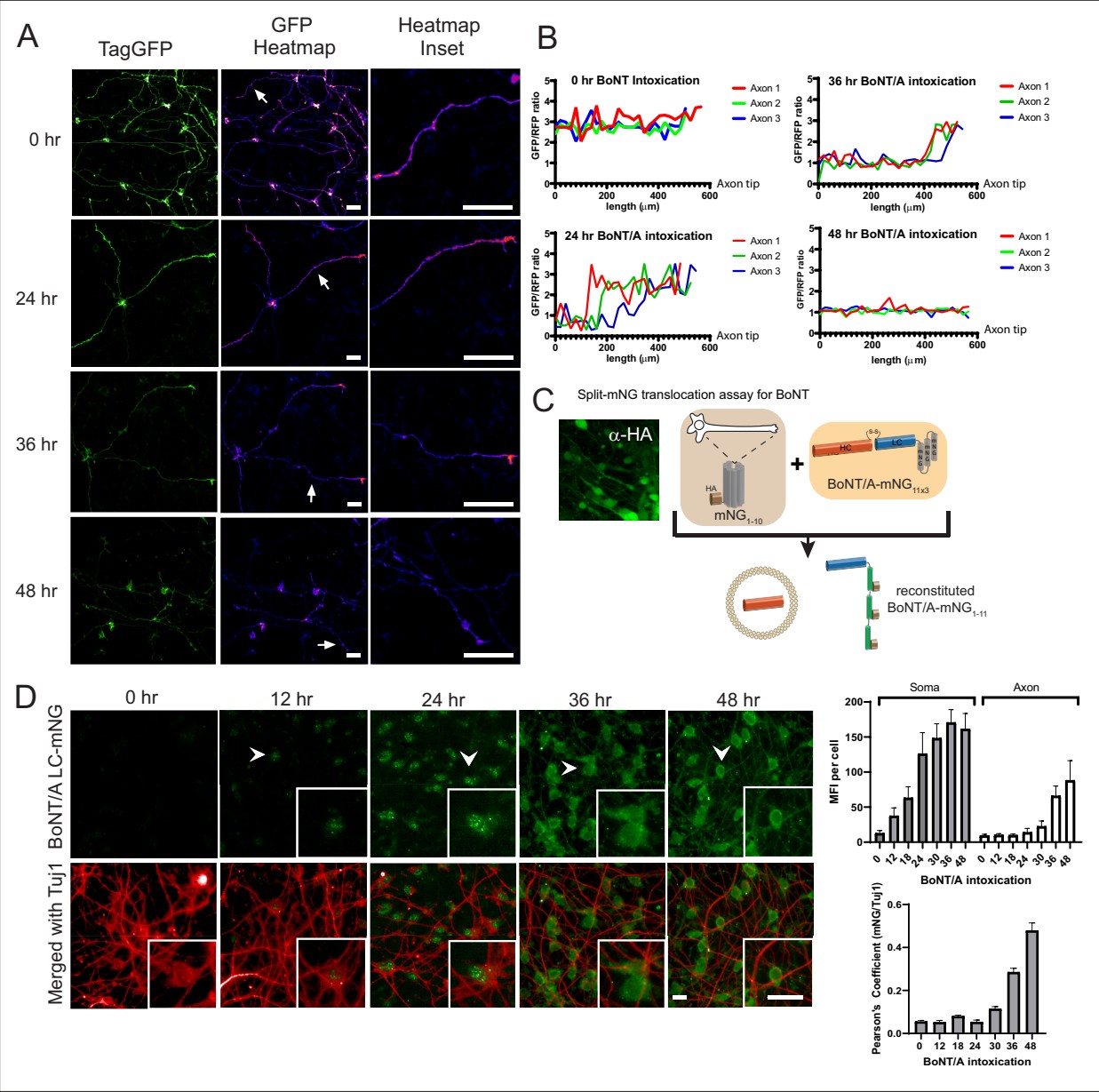

**Figure 2.** Botulinum neurotoxin A (BoNT/A) is first detected in the neuronal soma and then emanate to the axons. (**A**) Red SNAPR cells co-cultivated at ¼ with ¾ of unlabeled Ren-VM were imaged for 48 hr the after addition of BoNT/A. The GFP color-coded intensity signal is displayed in the second column. (**B**) Quantification of GFP/RFP signal along the length of individual neurites (Axon 1, 2, 3) at various time points. (**C**) Schematic diagram of split-mNG (NeonGreen Fluorescent Protein) detection system, consisting of ReNcell VM expressing cytosolic mNG$_{1-10}$ and mNG$_{11}$-tagged BoNT/A. Fluorescence occurs after binding of mNG$_{11}$-tagged BoNT/A to mNG$_{1-10}$. (**D**) Time-course of fluorescence reconstitution after exposure to mNG$_{11}$-tagged BoNT/A and quantification of Mean Fluorescence Intensity in the soma and neurites of cells. Scale bars 20 μm. Mean + SEM with n = 3 experiments with at least 20 cells from each experiment.

*Figure 3—figure supplement 1B*. The tight clustering of the controls demonstrates high reproducibility between experiments. The data was ranked to establish cut-offs for hits selection (*Figure 3E*).

Most hits (363) were genes required for BoNT/A intoxication (red dots); interestingly a significant fraction of hits (76) resulted in enhanced intoxication (blue dots) (*Figure 3F*). Many of the positive regulators resulted in higher rescue than the siTXNRD1 control, suggesting that previously unidentified molecular processes are critical for BoNT/A intoxication.

To exclude potential indirect effects, we first used nuclei counts and identified 289 genes that significantly affect neuronal survival (*Figure 3—figure supplement 1C*). 80 positive and eight

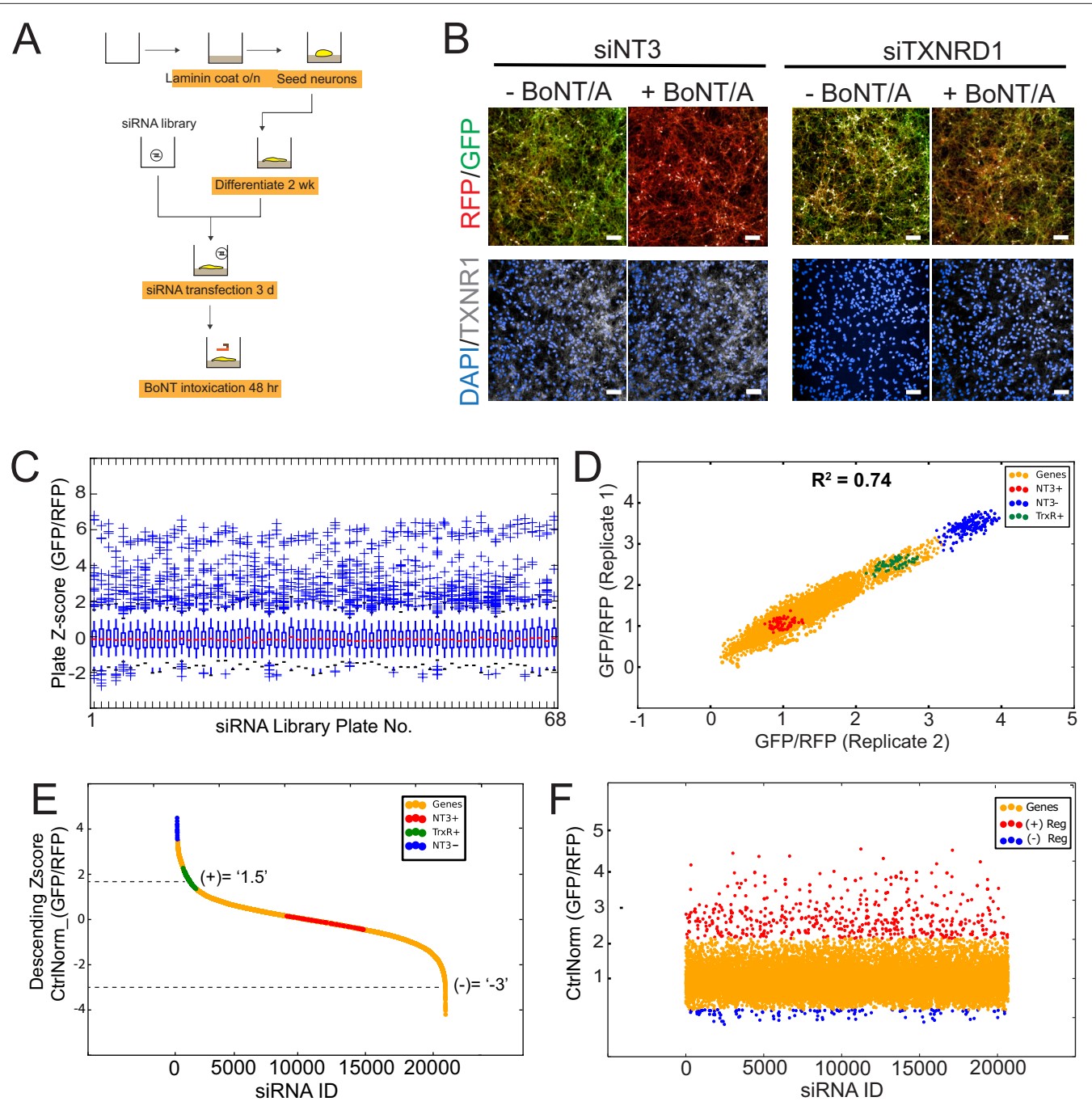

**Figure 3.** Genome-wide RNAi screen of Botulinum neurotoxin A (BoNT/A)-treated Red SNAPR cell line. (**A**) Schematic diagram of assay pipeline. (**B**) Cells treated with non-targeting control siRNA (siNT3) and positive control siRNA (siTXNRD1), then incubated with BoNT/A for 48 hr. Cells were stained with DAPI and TXNRD1 antibody post-fixation. Scale bars 50 µm. (**C**) ScreenSifter Z-score analysis of GFP/RFP ratio index of whole genome siRNA library plates. (**D**) R-squared analysis of both genome-wide RNAi screen replicates. Blue dots = No toxin control, Red dots = Toxin control, Green dots = Positive control (siTXNRD1). (**E**) Descending Z-score of control-normalized and averaged GFP/RFP ratio of genome-wide screen to determine cut-off values of +1.5 for positive regulators and –3 for negative regulators (Column 'Zscore_Log10_CtrlNorm' in **Figure 3—source data 1**). (**F**) Chronological order of genome-wide control-normalized screen reflecting the positive regulators (red) and negative regulators (blue) derived from (**E**).

The online version of this article includes the following source data and figure supplement(s) for figure 3:

**Source data 1.** Source data for **Figure 3** and **Figure 3—figure supplement 1** Excel sheet containing raw data of genome-wide screen in **Figure 3—figure supplement 1**.

**Figure supplement 1.** Genome-wide RNAi screen data normalization and refinement.

negative genes were sifted out of the hit list (*Figure 3—figure supplement 1D*). Next, we carried out a duplicate deconvoluted siRNA screen on the hit list to exclude potential off-target siRNAs (*Jackson and Linsley, 2010*; *Figure 3—figure supplement 1E*). Using this approach, 35 genes could not be confirmed by two independent siRNAs and thus were sifted out. To ensure the hit list only contains genes significantly expressed in neurons, we carried out RNA sequencing analysis on differentiated Red SNAPR cells (*Figure 3—figure supplement 1F*). A cut-off threshold of counts-per-million (CPM) at 0.25 ($\log_2$CPM = –2) was used (gray bars) and 31 more genes (18 positive, 13 negative) were excluded.

## The surface expression of BoNT/A receptor, SV2, is highly regulated

We next focused on genes influencing BoNT/A cell surface binding by studying the surface expression of SV2, the BoNT/A receptor. We incubated fixed and non-permeabilized cells with an antibody against the extracellular domain of SV2A to quantify cell surface exposure (*Figure 4A*). Depletion of VAMP2, a known regulator of synaptic vesicle fusion and SV2 trafficking (*Pennuto et al., 2003*), resulted in a 60% reduction of surface SV2 levels relative to non-targeting control (siNT3) (*Figure 4B*). By contrast, siTXNRD1 did not affect the SV2 surface signal (*Figure 4B*).

We screened a library of hits identified in *Figure 3F* and identified 105 genes that affect SV2, with 38 gene depletions reducing (Repressors, SV2(-) in table) and 67 gene depletions enhancing (Enhancers, SV2(+) in table) surface SV2 (*Figure 4C*, *Supplementary file 1*).

For instance, the depletion of the clathrin light chain (siCTLC), a known regulator of endocytosis, increased the surface staining of SV2 (*Yao et al., 2010*). By contrast, the depletion of Rab11b decreased surface SV2 which could be due to a block in the recycling of SV2 from endosomes to the cell surface (*Giorgini and Steinert, 2013*). Using STRINGS analysis on 'surface SV2 repressors', we identified a closely associated network of genes related to clathrin-mediated endocytosis such as AP2M1, CTLC, and TDRD1. In addition, distinct subunits of the V-ATPase were identified which can function as adaptins to facilitate endocytosis (*Geyer et al., 2002*; *Figure 4D*). The majority of the surface SV2 enhancers are associated with G-protein signaling, which controls neuronal excitation (*Figure 4E*). This agrees with earlier findings (*Figure 1D*) where increased neuronal activity favors intoxication as observed when $K^+/Ca^+$ are spiked in the medium. Endocytic signaling and membrane trafficking gene families such as Rabs, Arfs, and SNAREs were also revealed, congruent with a role in SV2 exocytosis (*Figure 4E*).

## Network analysis reveals regulators of signaling, membrane trafficking and thioreductase redox state involved in BoNT/A intoxication

Among the positive regulators of the screen, 135 hits did not influence significantly surface SV2 levels and are thus likely to function in post-endocytic processes (*Supplementary file 2*). However, we cannot formally exclude that they could affect the binding of BoNT/A to the cell surface independently of SV2. 92 positive regulators (required for intoxication, in red) and 43 negative regulators (reducing intoxication, blue) were mapped to their intracellular localities (*Figure 5A*). Several gene products were localized to endosomes and 16 were associated with the Golgi and ER. At the Golgi, one hit was the glycosylation enzyme B4GALT4, known to be involved in the biosynthesis of the ganglioside co-receptor of BoNT/A, GT$_{1b}$. Using the STRING database, a protein-protein interaction network of hits was generated (*Figure 5B*; *Szklarczyk et al., 2021*). A subnetwork was constituted of heat-shock protein (HSP) chaperones of the HSP70 family (HSPA4 and HSPA14). Another chaperone, HSP90, has been linked to the translocation of the clostridial toxins C2 toxin and BoNT/A into the cytosol (*Azarnia Tehran et al., 2017*; *Haug et al., 2003*). As HSP90 and HSP70 act in a sequential cascade for protein folding (*Genest et al., 2019*), our results suggest that such a chaperone cascade might help BoNT/A LC refold in the cytosol after translocation.

A large subnetwork of signaling and GPCR-related genes was identified, centered on the SRC tyrosine kinase (*Figure 5B*). Depletion of SRC resulted in one of the most stringent intoxication blocks (*Figure 5C*). Interestingly, the tyrosine phosphatase PTPN6 was also one of the strongest negative regulators (*Chong and Maiese, 2007*; *Figure 5C*). It has been proposed that SRC directly phosphorylates and regulates the BoNT LC catalytic domain (*Ferrer-Montiel et al., 1996*; *Ibañez et al., 2004*; *Kiris et al., 2015*). On the other hand, SRC and its partners are crucial for retrograde membrane trafficking and might influence post-endocytic BoNT/A trafficking (*Chia et al., 2021*; *Sandilands and*

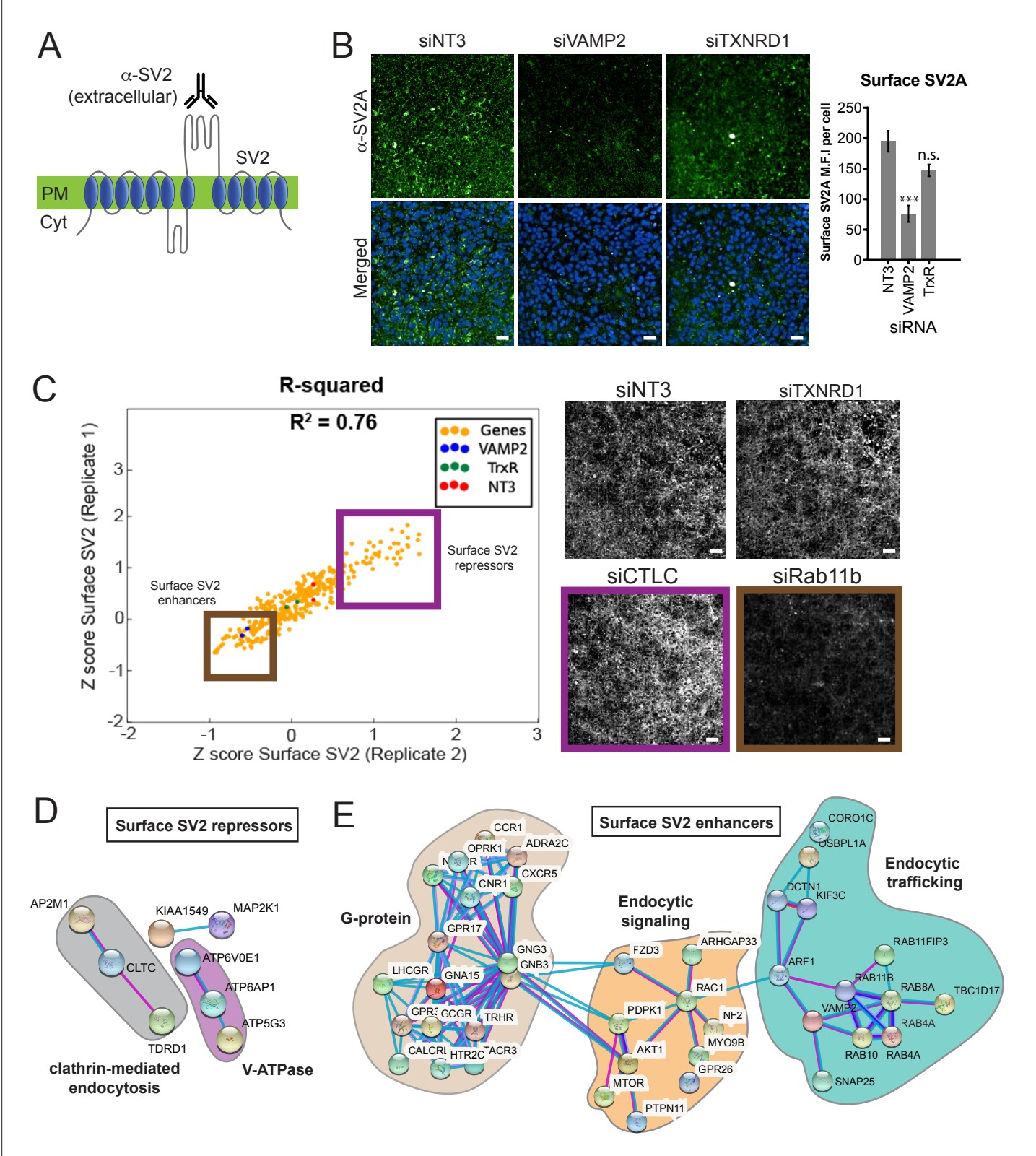

**Figure 4.** Surface expression of the Botulinum neurotoxin A (BoNT/A) receptor SV2 is regulated by endocytic trafficking and signaling genes, forming a cohort of the positive regulators of the genome-wide screen. (**A**) Detecting surface synaptic vesicle protein 2 (SV2) using a specific antibody against the extracellular loop of SV2. (**B**) siVAMP2 as a positive control for surface SV2 expression. ReNcell VM treated with siNT3, siVAMP2, and siTXNRD1 for 3 days and stained with SV2A antibody after fixation without membrane permeabilization. Quantification of surface SV2 per cell where whole image

*Figure 4 continued on next page*

*Figure 4 continued*

SV2A MFI was divided by total nuclei count (DAPI). Scale bars 50 µm. Mean + SEM with n = 3 experiments with at least 200 cells from each experiment. (**C**) Replicate screen results for surface SV2 regulators from genome-wide positive hits. Purple box = surface SV2 repressors e.g., siCTLC (clathrin light chain); Brown box = surface SV2 enhancers e.g., siRab11b. siNT3 and siTXNRD1 does not regulate surface levels of SV2. Scale bars 50 µm. (**D**) STRINGS analysis of surface SV2 repressors. (**E**) STRINGS analysis of surface SV2 enhancers.

*Frame, 2008*). The screen identified TXNRD1, consistent with previous literature, but also several genes involved in thioredoxin reduction, including Methionine sulfoxide reductase A (MsrA), Thioredoxin domain-containing protein 17 (TXNDC17), Nucleoredoxin (NXN) and Selenoprotein N (SEPN1) (*Figure 5D*). It suggests that these proteins either interact directly with BoNT/A or are required for TXNRD1 function.

A set of genes linked to the translocon (SEC61G, SEC61B, TRAM1, SERP1) were identified (*Figure 5E*). GO: MF analysis (Molecular Function) indicates a high enrichment level of these ER translocon genes (*Figure 5—figure supplement 1A*). The requirement of the translocon suggests its involvement in the translocation of the toxin to the cytosol as is the case for other toxins (*Moreau et al., 2011*; *Nowakowska-Gołacka et al., 2019*). However, this hypothesis implies that the toxin travels from endocytic vesicles at its site of internalization to ER membranes where the translocon is localized.

Interestingly, genes for the retromer were also identified, VPS35, VPS26A, SNX1, and SNX27, with a high level of enrichment in gene ontology analysis (*Figure 5F*, *Figure 5—figure supplement 1B*). These genes did not significantly affect surface SV2 levels. The retromer has been linked to endosomes to Golgi traffic, suggesting the toxin might need to traffic between these organelles.

## Retro-axonal traffic of the BoNT/A receptor SV2 requires the retromer

We next analyzed the effect of VPS35 depletion on the kinetics of BoNT/A-mNG$_{11}$ arrival in the cytosol. The control siNT3-treated cells showed a progressive increase in signal from 12 to 48 hr, with a progressive appearance of signal in the neurites (*Figure 6A*). By contrast, siVPS35-treated cells only displayed some soma-localized signal after 48 hr (*Figure 6A*).

To test directly the trafficking of BoNT/A intracellularly proved difficult as the reagents we tested were not sensitive enough to image the trafficking of the toxin. Since BoNT/A binds to SV2 at the cell surface, we wondered whether the receptor itself was trafficked retrogradely in neurites. We generated a ReNcell VM cell line expressing a chimera of SV2 with the Dendra fluorescent protein. We found that Dendra-SV2 is distributed in the whole neuron, similarly to the endogenous protein (*Figure 6B*).

Next, we depleted cells of VPS35 and found that Dendra-SV2 accumulated in bulbous structures at the neurite tips. To quantify this phenomenon, we compared the number and size of SV2 puncta in neurites in control and siVPS35-treated cells. Using ImageJ to threshold and select for particles of interest, we measured the number of particles along every 50 µm segment of the neurite, starting from the neurite tips (*Figure 6C*). In control cells, SV2 punctas were homogeneously spread throughout the neurites. However, in VPS35-depleted cells, SV2 punctas were enriched at the neurite tip (1–50 µm) and were significantly reduced with the rest of the neurite (*Figure 6D*). These puncta were also much larger than in control cells. This evidence indicates that SV2 is trafficked retro-axonally in a retromer-dependent fashion, thus consistent with the notion of BoNT/A retrogradaly to the neuronal body bound to its receptor.

## BoNT/A trafficks through the Golgi apparatus

The implication of the retromer suggested that BoNT/A transits through the Golgi apparatus. To test this hypothesis, we sought a method to detect BoNT/A in different intracellular compartments. We decided to exploit the split-monomeric Neon Green (mNG) fluorescence reconstitution approach (*Luong et al., 2020*). This approach relies on having the 10 beta strands of Neon Green fluorescent protein (mNG$_{1-10}$) and the 11th beta-strand in different proteins. When the two proteins can interact, fluorescence is reconstituted. We generated a ReNcell VM cell line stably expressing mNG$_{1-10}$ fused with an HA tag and a fragment of β1,4-galactosyltransferase 1 for targeting to the Golgi (Golgi-mNG$_{1-10}$) (*Luong et al., 2020*; *Figure 7A*). We verified by immunofluorescence that the construct was

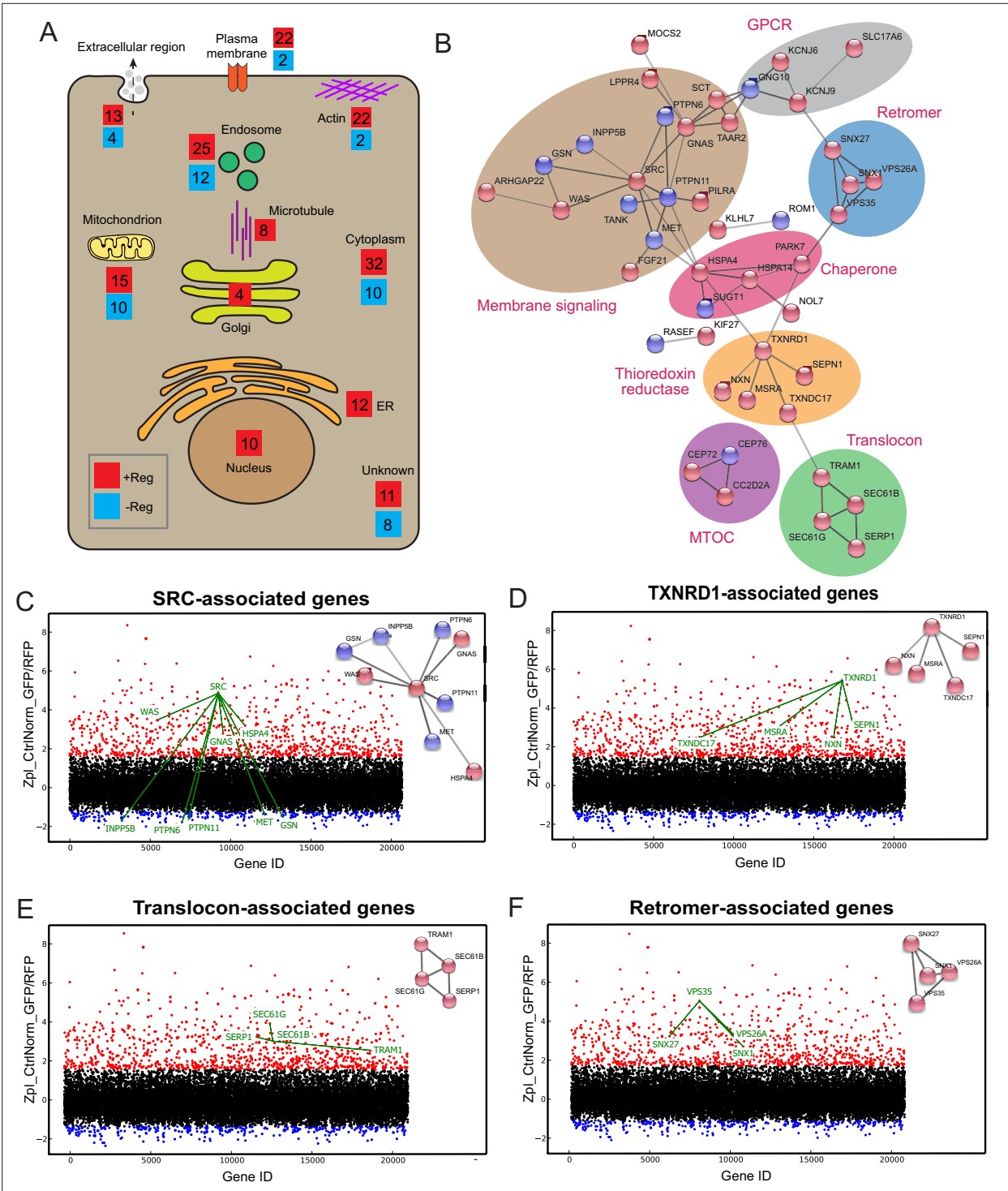

**Figure 5.** Protein network analysis of genome-wide hits reveals known and novel regulators of Botulinum neurotoxin A (BoNT/A) intoxication. (**A**) Summary diagram of all positive and negative hits mapped to their intracellular localities. (**B**) STRINGS network analysis of connected hits (non-connected hits are excluded). Associated genes tied to respective cellular molecular complexes/processes are bounded and annotated. ScreenSifter analysis revealed (**C**) Src-associated genes (**D**) TXNRD1-associated genes (**E**) Translocon-associated genes (**F**) Retromer-associated genes.

The online version of this article includes the following figure supplement(s) for figure 5:

**Figure supplement 1.** Gene ontological analysis of positive hits from genome-wide intoxication screen.

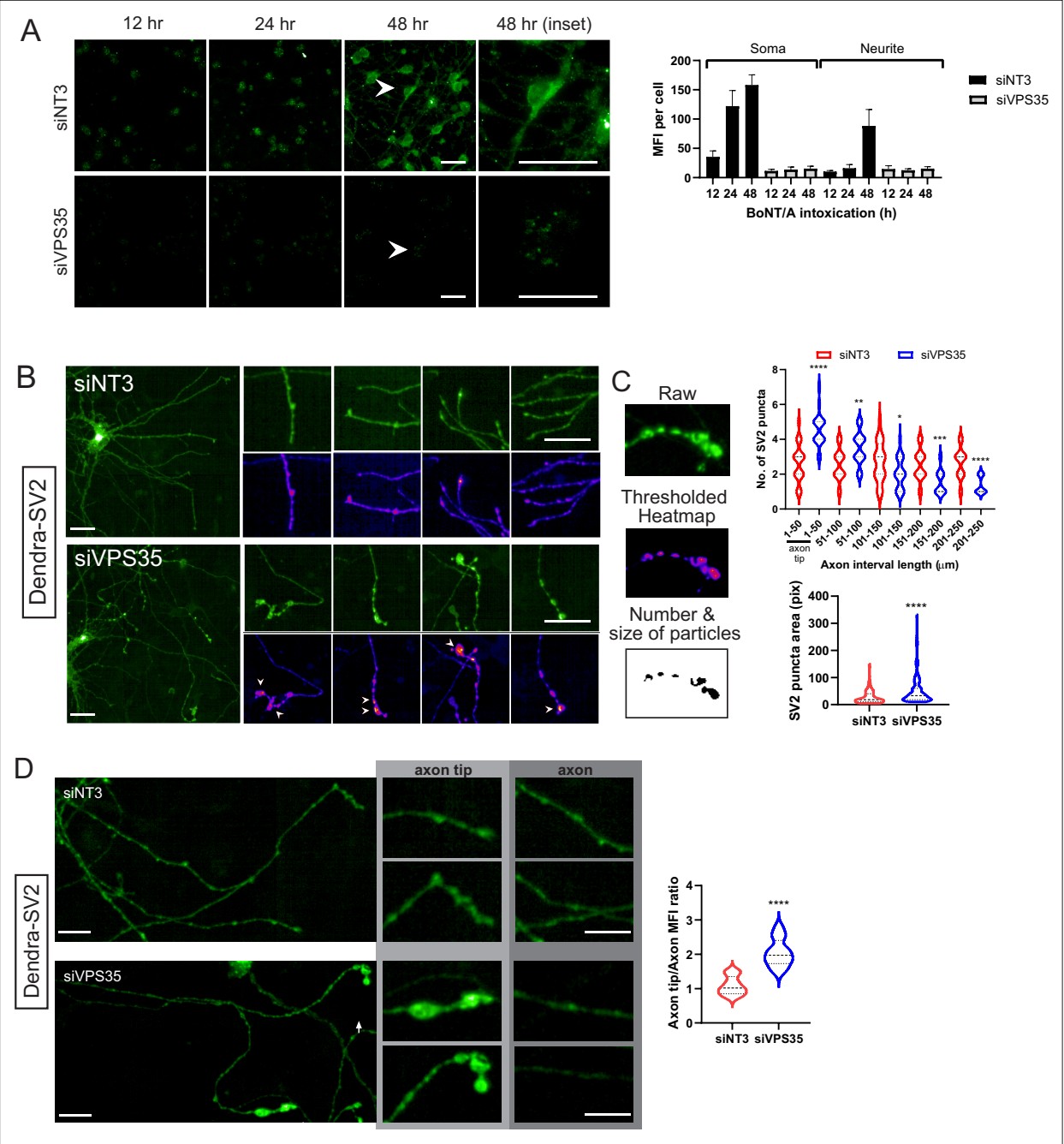

**Figure 6.** Retromer is required for retro-axonal trafficking of the Botulinum neurotoxin A (BoNT/A) receptor. (**A**) Cytosolic mNG$_{1-10}$ expressing RenVM cells were incubated with 50 nM of BoNT/A-mNG$_{11}$ and fixed at 0, 24, 36, and 48 hr time points. The mean fluorescence intensity (MFI) was quantified in 20 cells for each condition and time point, in the soma and neurites. Scale bar 20 µm. (**B**) ReNcell VM stably expressing Dendra-synaptic vesicle protein 2 (SV2) treated with siVPS35 for 3 days and imaged after fixation. Insets show axons of interest from each condition and arrowheads indicate enlarged SV2 puncta. Scale bar 20 µm and inset shown at 10 µm. (**C**) Method for quantifying the number and size of SV2 puncta. tGFP image is background-subtracted, thresholded then analyzed for the number and size of particles. Quantification of the number of SV2 puncta along every 50 µm segment of the axon starting from the axon tip from (**B**) Quantification of area of SV2 puncta in whole axons from B. (**D**) Comparison of MFIs at axons and axon tips from control and siVPS35-treated cells. Quantification of axon tip/axon MFIs. Axon analysis graphs are obtained from at least 30 axons across three independent experiments. Scale bar 10 µm.

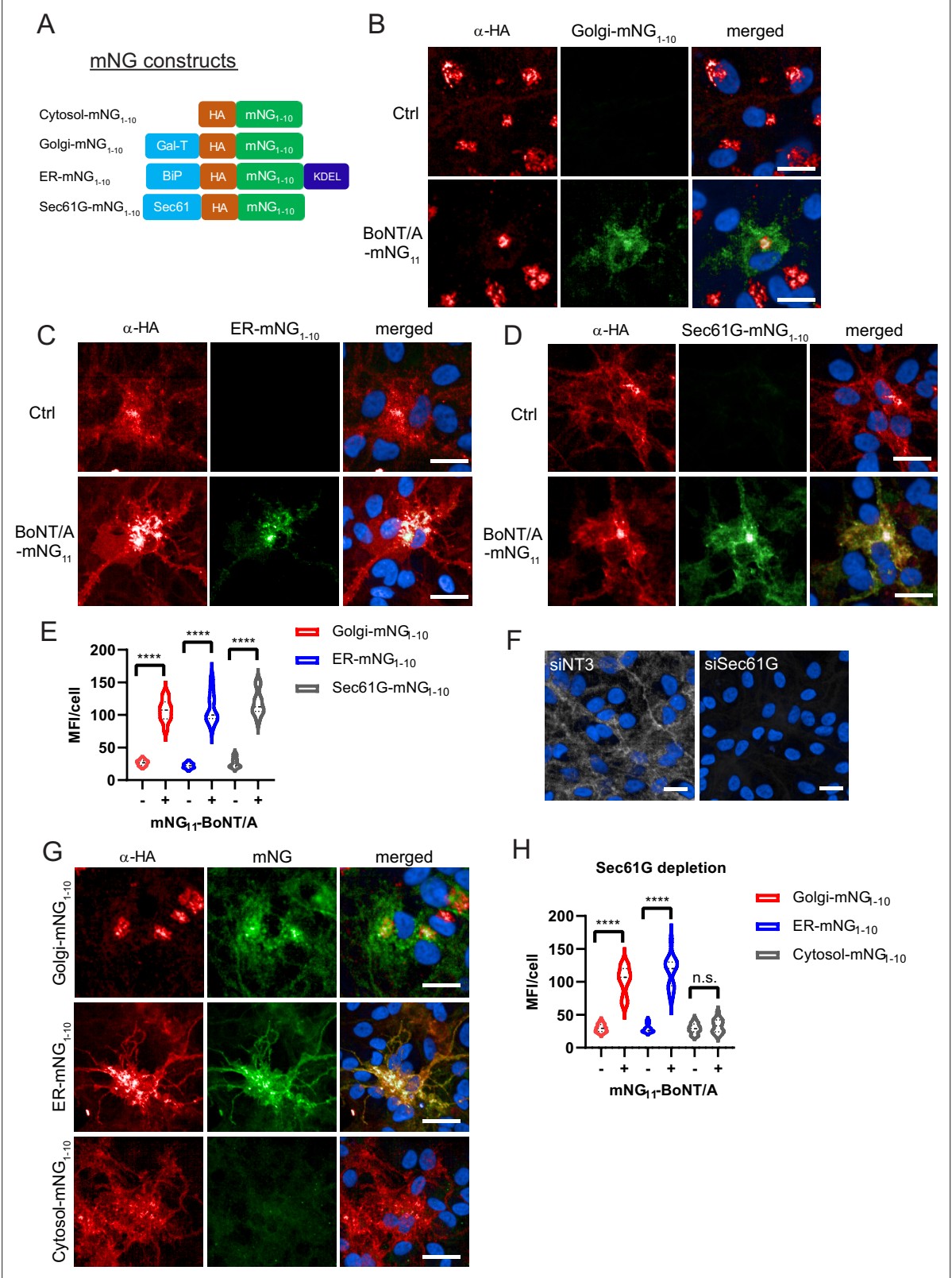

**Figure 7.** Botulinum neurotoxin A (BoNT/A) is retrogradely trafficked through Golgi and ER membranes as revealed by split-GFP reconstitution. (**A**) Split monomeric NeonGreen (Split-mNG) constructs targeted to cytosol, Golgi, ER lumen (KDEL sequence), and ER membranes (Sec61 transmembrane domain) were stably expressed in ReNcell VM and incubated with 50 nM of BoNT/A-mNG$_{11}$. Control cells (without BoNT/A-mNG$_{11}$) are shown at top panels of each condition (**B**) Cells expressing Golgi-mNG$_{1-10}$ and showing reconstituted mNG fluorescence. Overlap of HA antibody and mNG

*Figure 7 continued on next page*

*Figure 7 continued*

signals in the Golgi (**C**) ER-mNG$_{1-10}$ expressing cells showing mNG fluorescence and distinct overlap of HA antibody and mNG in the ER. (**D**) Sec61G-mNG$_{1-10}$ expressing cells showing mNG fluorescence distinct overlap of HA antibody and Sec61G signals. (**E**) Quantification of mean fluorescence intensities (MFI)/cell from (**B–D**) showing increased mNG fluorescence in Golgi, ER, and Sec61G-expressing cells after addition of BoNT/A-mNG11. (**F**) Representative image of SEC61G depletion in ReNcell VM. (**G**) Golgi-mNG$_{1-10}$, ER-mNG$_{1-10}$, and Cytosol-mNG$_{1-10}$-expressing cells are depleted with siSEC61G and incubated with BoNT/A-mNG$_{11}$ for 48 hr. (**H**) Quantification of MFI/cell from (**G**). Graphs are obtained from at least 200 cells across three independent experiments. Scale bars 20 μm.

strictly localized at the Golgi by the HA antibody staining and that no GFP fluorescence was detectable (*Figure 7B*).

Upon incubation for 48 hr with BoNT/A-mNG$_{11}$, we observed reconstitution of mNG fluorescence at the Golgi (*Figure 7B*). In addition, there was mNG fluorescence in the cell soma with a reticulated pattern, which could correspond to the ER. Interestingly, the HA pattern was also partially looking like ER in these cells. Thus the reconstituted mNG was not only restricted to the Golgi, which could be due to retrograde trafficking after complementation. Nonetheless, the Golgi pattern clearly indicates that BoNT/A traverses this organelle before translocation. In ReNcell VM cells, the Golgi apparatus is located almost exclusively in the soma of neurons.

## BoNT/A trafficks to the ER and translocates through the Sec61 complex

To test whether BoNT/A can reach the ER, we next developed a chimera of mNG$_{1-10}$ localized to the ER, based on the soluble, ER-resident BiP protein (ER-mNG$_{1-10}$) (*Luong et al., 2020*). As for the Golgi construct, there was no fluorescence in the absence of BoNT/A and the construct had the typical pattern of an ER protein (*Figure 7C*). 48 hr after intoxication with BoNT/A-mNG$_{11}$, we could detect mNG in the ER (*Figure 7C*). To further confirm that BoNT/A can reach a Sec61-enriched compartment, we also developed ReNcell VM cell lines stably expressing a split-mNG$_{1-10}$ construct fused with the transmembrane protein Sec61G (*Figure 7A*). As for the BiP-based construct, the Sec61 staining pattern before the addition of the toxin was consistent with the ER. After intoxication, the reconstituted mNG fluorescence had a typical ER pattern (*Figure 7D*). Quantification of fluorescence per cell confirmed that the three reporters were equally accessible to BoNT/A, indicating that the toxin traffics through the Golgi and ER in neurons (*Figure 7E*).

Next, to test whether Sec61 is involved in BoNT/A LC translocation, Golgi-mNG$_{1-10}$, ER-mNG$_{1-10}$, and Cytosol-mNG$_{1-10}$-expressing cells were depleted of Sec61G with siRNA (*Figure 7F*). When incubated with BoNT/A-mNG$_{11}$, the reconstituted mNG fluorescence appeared in Golgi-mNG$_{1-10}$ and ER-mNG$_{1-10}$ cells, while there was a striking reduction of fluorescence in cytosolic mNG$_{1-10}$ expressing cells (*Figure 7G and H*). These results indicate that loss of Sec61G does not prevent BoNT/A uptake nor its trafficking to Golgi and ER compartments, but is instrumental to the translocation of BoNT/A LC into the neuronal cytosol at the soma.

Overall, these reporters reveal the complexity of BoNT/A trafficking in neurons: after internalization at active synapses in neurites, the toxin is transported retrogradely to the Golgi, then the ER before being translocated to the cytosol.

## Discussion

To improve the understanding of BoNT/A intoxication, we designed the engineered BoNT reporter cell line, Red SNAPR. The line is easily maintained, highly sensitive to the toxin and the assay is a direct readout of residual GFP fluorescence after toxin cleavage. The assay encompasses all steps of BoNT/A intoxication from binding and endocytosis to light chain translocation and substrate cleavage. While FRET-based reporter assays of BoNT have been established, they tend to be limited by photobleaching and spectral bleedthrough from donor/acceptor fluorescence. Moreover, FRET efficiency is drastically reduced in fixed cells (*Anikovsky et al., 2008*; *Dong et al., 2004*). By contrast, SNAPR cleavage is quantified through the ratio of RFP to GFP signal, thanks to the instability of the GFP moiety after cleavage, probably due to the N-terminal arginine residue generated by BoNT/A cleavage at Q$^{197}$R (*Tasaki et al., 2012*). As noted by one reviewer, the assay may be sensitive to perturbation in the general rate of protein degradation, a consideration to

keep in mind when evaluating the results of large-scale screens. On the other hand, this convenient and simple assay could replace animal-based assays that are used to measure BoNT potency. In addition, SNAPR could be implemented in transgenic mice to study the process of BoNT/A inhibition in vivo.

The sensitivity of the ReNcell VM cells to BoNT/A is highly enhanced by treatments that favor the formation of active synapses, consistent with the fact that BoNT/A binds the synaptic protein SV2, which only becomes exposed at the surface when neurons fire synaptic vesicles. Consistently, a hundred genes regulating BoNT/A intoxication (a quarter of the hits) actually impact cell surface exposure of SV2. The genes identified are involved in endocytic coupling, G protein-coupled receptors, and other signaling genes. Thus, it seems highly likely that in our experimental model, BoNT/A is internalized at the tip of neurites (with axonal properties).

After binding to SV2 and internalization, BoNT/A LC does not appear to translocate in the synaptic bouton as the reporter is unperturbed. The spatiotemporal pattern of BoNT/A activity and the cytosolic detection of the protein both indicate that the toxin appears in the soma of neurons about 12 hr after the start of incubation. Presumably in our cell culture system, these 12 hr are required for BoNT/A trafficking from synapses to the soma's cytosol. Progression of cleavage of the SNAPR reporter from the soma to the neurite's terminal requires an additional 36 hr. The split-GFP organellar reporters indicate that BoNT/A traffics from endosomes to Golgi and then to the ER.

The genetic signature of BoNT/A intoxication requirements is consistent with this trafficking route. For instance, a significant positive regulator is the tyrosine kinase SRC. We and others have reported on the role of SRC at the Golgi/ER interface (*Pulvirenti et al., 2008*; *Weller et al., 2010*). We previously reported SRC's role in regulating Pseudomonas Exotoxin A (PE) trafficking between Golgi and ER (*Bard et al., 2003*). More recently, we have shown the role of Src in directly controlling GBF1, a GTP Exchange Factor involved in Golgi to ER traffic (*Chia et al., 2021*). Interestingly, the counteracting tyrosine phosphatases PTPN6 and PTPN11 are negative regulators of BoNT/A, and their depletion favors intoxication (*Frank et al., 2004*; *Somani et al., 1997*). To note, SRC has also been proposed to act directly on the toxin to activate it (*Ferrer-Montiel et al., 1996*; *Ibañez et al., 2004*; *Kiris et al., 2015*).

These data suggest that BoNT/A effects could be counteracted clinically by Src targeting drugs. This is important as serious or even lethal botulinic intoxications, while rare, still occur in developed countries like France (*Cas de botulisme alimentaire à Bordeaux, 2023*).

Another four genes are linked to the retromer complex, which controls endosomes to Golgi traffic. For BoNT/A, traffic from endocytic vesicle to Golgi requires retro-axonal transport, which is supported by the requirement for dynein. Previous studies had shown retro-axonal transport for BoNT/A (*Harper et al., 2016*; *Restani et al., 2012*). By extension, these results suggest that SV2 is also transported retro-axonally (*Lund et al., 2021*; *Tanner et al., 1996*). Consistently, we found that SV2 accumulates at neurite tips when the retromer complex is silenced. SV2 is implicated in epilepsy and various neurodegenerative diseases such as Alzheimer's and Parkinson's (*Ciruelas et al., 2019*; *Stout et al., 2019*).

Another complex required is the translocon. When Sec61G is depleted, mNG-complementing BoNT/A still accumulates in the ER but is unable to reach the cytosol, indicating that the Sec61 complex is involved in retrotranslocation. While the best-described function is the co-translational insertion of proteins in the ER, Sec61 has also been implicated in retrotranslocation linked to ER-associated degradation (ERAD) (*Römisch, 2017*). Sec61 thus likely functions as a bidirectional channel for proteins (*Römisch, 2017*).

Overall, we propose that the BoNT/A-SV2 complex is internalized by ReNcell VM cells at the neurite tip where SV2 is exposed. While we did not demonstrate this point formally in RenCell VM cells, it is well established in other neurons and it is consistent with the increased toxin sensitivity after neuronal stimulation (*Figure 8A*). After internalization, instead of being translocated from endocytic vesicles, the toxin is retrogradely trafficked to the soma-located Golgi in a retromer-dependent fashion (*Figure 8B*). After trafficking through the Golgi, the toxin is transported to the ER (*Figure 8C*). There, BoNT/A LC translocates via the Sec61 translocon into the cytosol. It is likely that LC detachment from ER membranes requires the separation of heavy and light chains through the activity of thioredoxin reductases and chaperones (*Figure 8C*). Thus, in neurons in culture, BoNT/A LC likely cleaves SNAP25 first in the soma before reaching the neurite terminals where it blocks the fusion of synaptic vesicles at the plasma membrane (*Figure 8D*). The location of the Golgi apparatus virtually exclusively in the

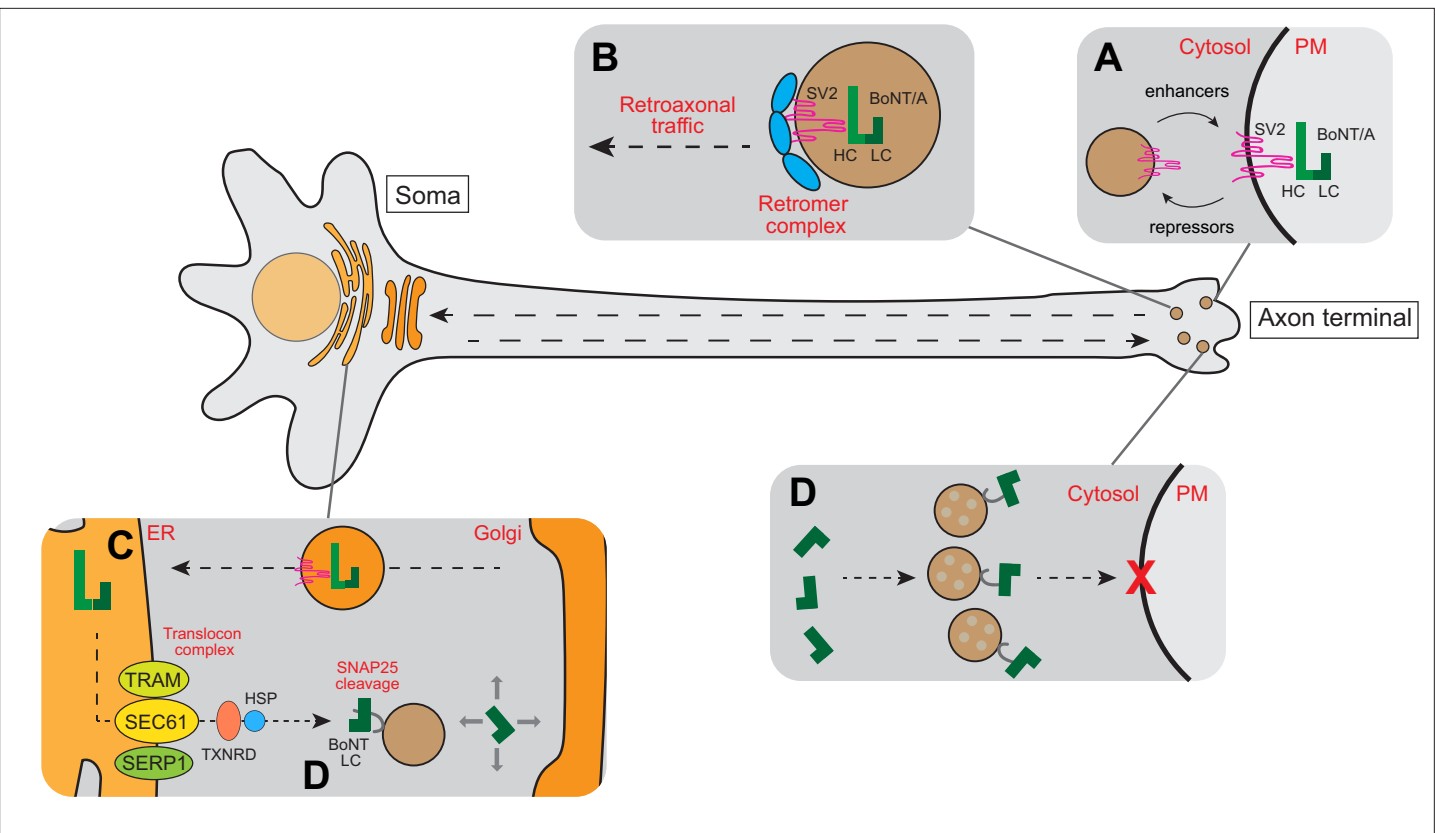

**Figure 8.** Schematic model for intracellular trafficking of BoNT/A. (**A**) BoNT/A HC binds to its cognate receptor SV2. Surface expression of SV2 is regulated by a cohort of enhancers and repressors. (**B**) BoNT/A-containing endosomes are retro-axonally trafficked to the soma through the retrograde function of the retromer complex. (**C**) BoNT/A eventually traffics to the translocation-competent ER via the Golgi. (**D**) The ER SEC61 translocon complex facilitates LC translocation of BoNT/A from the ER lumen into the cytosol where the TXNRD1 and HSP complexes release and refold BoNT/A LC. The LC diffuses and cleaves SNAP25 in the soma. (**E**) Progressive diffusion of LC into the axons and axon terminals ultimately cleaves SNAP25 on neurotransmitter-containing vesicles, blocking neurotransmission at the synapse.

soma of differentiated Ren-VM cells explains why the toxin has to traffic to the soma. This trafficking in turn explains the 48 hr between the addition of the toxin and the full cleavage of SNAPR.

Our study contradicts the long-established model of BoNT intoxication, which is described in several reviews specifically dedicated to the subject (*Dong et al., 2019*; *Pirazzini et al., 2017*; *Pirazzini et al., 2016*; *Rossetto et al., 2021*). In short, these reviews support the notion that BoNTs are molecular machines able to mediate their own translocation across membranes. This notion has convinced some cell biologists interested in toxins and retrograde membrane traffic, who follow this model of BoNT mode of translocation in their reviews (*Mesquita et al., 2020*; *Williams and Tsai, 2016*).

But is this notion well supported by data? A careful examination of the primary literature reveals that early studies indeed report that BoNTs form ion channels at low pH values (*Donovan and Middlebrook, 1986*; *Hoch et al., 1985*). These studies have been extended by the use of patch-clamp (*Fischer et al., 2009*; *Fischer and Montal, 2007a*). These works and others lead to various suppositions on how the toxin forms a channel and translocate the LC (*Fischer and Montal, 2007b*; *Pirazzini et al., 2016*).

However, only a single study claims to reconstitute in vitro the translocation of BoNT LC across membranes (*Koriazova and Montal, 2003*). In this paper, the authors report one key experiment using a system of artificial membranes separating two aqueous compartments (*Figure 3D*, *Koriazova and Montal, 2003*). They load the toxin in the cis compartment and measure the protease activity in the trans compartment after incubation. However, when the experimental conditions described are actually converted in terms of molarity, it appears that the cis compartment was loaded at $10^{-8}$ M BoNT and that the reported translocated protease activity is equivalent to $10^{-17}$ M (*Figure 3D*, *Koriazova and Montal, 2003*). Thus, in this experiment, about 1 LC molecule in 100 million has

crossed the membrane. Such an extremely low transfer rate does not tally with the extreme efficiency of intoxication in vivo, even while taking into account the difference between artificial and biological membranes.

In sum, a careful analysis of the primary literature indicates that while there is ample evidence that BoNTs have the ability to affect membranes and possibly create ion channels, there is actually no credible evidence that these channels can mediate the translocation of the LC. As mentioned earlier, it is unclear how such a self-translocation mechanism would function thermodynamically. It is worth noting that a similar self-translocation model was proposed for other protein toxins such as Pseudomonas exotoxin, which have a similar molecular organization as BoNT (*Taupiac et al., 1999*). However, it has since been demonstrated that the PE toxins require cellular machinery, in particular in the ER, for intoxication (*Bassik et al., 2013*; *Moreau et al., 2011*; *Schäuble et al., 2014*).

By contrast, our model proposes a mechanism without a thermodynamic problem, is consistent with current knowledge about other protein toxins, such as PE, Shiga and Ricin, and can help explain previously puzzling features of BoNT effects.

## Materials and methods
### Material availability and commercial reagents
The reagents used in this study are summarized in *Figure 9A*. BoNT/A was synthesized by IPSEN scientists as previously described (*Stancombe et al., 2012*). The cell lines and DNA constructs are available from IMCB (@ F. Tay) upon request.

### Constructs
Constructs used in this study are depicted in *Figure 9B* and were generated using gene synthesis and cloned into the pDONR221 entry vector (GeneArt, Thermo Fisher Scientific). The entry clones were subcloned into pLenti6.3-V5 destination vectors using gateway LR cloning (ThermoFisher Scientific). In the split-mNG fluorescence reconstitution system, the first 10 β-barrel helices of monomeric Neon Green (mNG) are fused with or without target proteins while the last beta-strand is fused with the protein of interest. Interaction of the two partners will reconstitute fluorescence (*Luong et al., 2020*). BoNT/A-mNG$_{11x3}$ was synthesized by the addition of 3 mNG$_{11}$ tags flanked with GSGSG (Gly-Ser) linkers at the N-termini of BoNT/A. siGENOME Human siRNA libraries were purchased from Dharmacon (Horizon Discovery).

### Cell lines
ReNcellVM human neural progenitor cell line (Sigma Aldrich) was maintained in ReNcell NSC Maintenance medium supplemented with 20 ng/mL of epidermal growth factor, EGF and basic fibroblast growth factor, bFGF at 37 °C with 5% $CO_2$. Cells were differentiated in differentiation medium (maintenance medium without EGF and bFGF); and addition of 10 ng/mL of both glial-derived nerve factor (GDNF) and brain-derived nerve factor (BDNF) for 2 weeks with a change of media every 3 days. Cells were seeded on culture flasks or plates precoated overnight at 4 °C with 20 ug/mL laminin in DPBS. The cells were subcultured approximately every 5 days (90% confluence) by detaching them with Accutase. Mycoplasma testing was conducted every other week to insure all experiments were conducted with mycoplasma-free cells.

### Stable cell line generation
Stable cell lines were generated using ViraPower Lentiviral Packaging Mix together with the following lentiviral constructs containing SNAP25 were cotransfected into HEK293FT cells using Lipofectamine 3000 reagent. Following the incubation of cells, a supernatant containing lentivirus was harvested and cellular debris was removed by centrifugation. The virus was transduced in ReNcell VM with polybrene reagent (8 mg/ml) and removed after 24 hr. Fresh complete growth medium was added and transduced cells were sorted using fluorescence-activated cell sorting using tGFP signal and selected using 10 ug/mL blasticidin in maintenance medium.

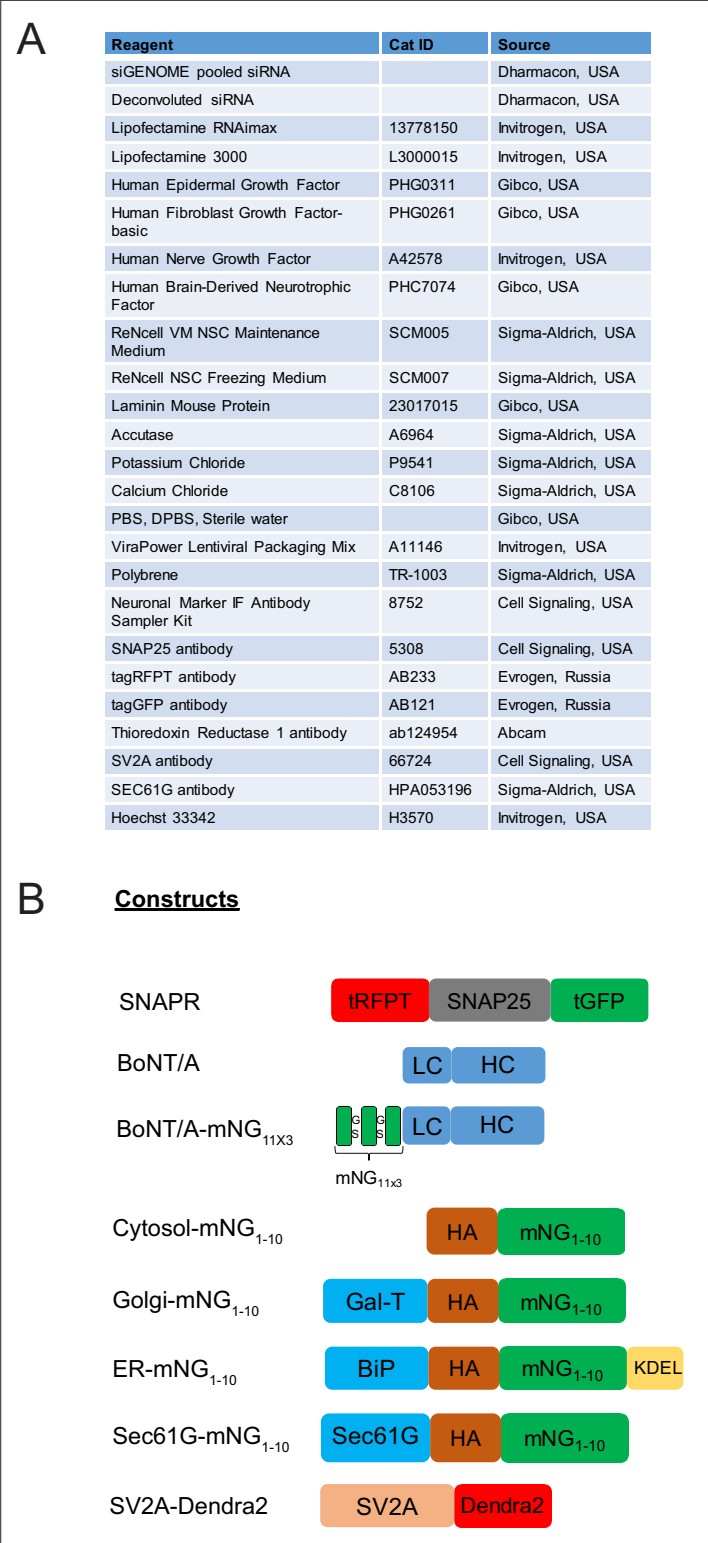

**Figure 9.** Reagents and constructs used in this study. (**A**) Reagents used in this study. (**B**) Genetic constructs used in this study.

### High-throughput siRNA and BoNT/A intoxication screen assay

The siRNA library was spotted in 384-well black µClear plates (Greiner). siRNAs were transiently transfected at a final concentration of 25 nM per well using 0.25 µl Lipofectamine RNAiMAX reagent in 7.25 µl of OptiMEM medium, according to the manufacturer's protocol. siNT3 (Non-targeting 3) was chosen as a control siRNA due to its least toxic properties in ReNcell VM. After 20 min of complex formation, complexes were dispensed to the differentiated cells per well using a Multidrop Combi dispenser (Thermo Fisher Scientific). Medium was removed at 72 hr post-transfection using Integra VIAFLO 384 (Integra Biosciences AG, Switzerland), and cells were washed with phosphate-buffered saline (PBS) twice before incubated with 6 nM BoNT/A in differentiation medium with 58 mM KCl and 2.2 mM CaCl2 for 48 hr. siRNA screening was performed in duplicate.

### Immunofluoresence

After 48 hr of BoNT/A intoxication, cells were fixed with 4% paraformaldehyde and 2% sucrose in PBS for 30 min and permeabilized with 0.2% Triton X-100 for a further 10 min. The cells were then stained with primary antibody diluted in 2%FBS in PBS for 2 hr. Cells were subsequently washed three times for 5 min with 2%FBS in PBS and stained for 1 hr with a secondary antibody conjugated with fluorophore and Hoechst 33342 diluted in 2% FBS in PBS. The cells were then washed three times for 5 min with PBS before imaging. We tested the following antibodies to follow BonT intracellular trafficking: BoNT/A LC polyclonal: R&D Systems (AF44839), BoNT/A LC monoclonal: R&D systems (MAB4489), BoNT/A polyclonal: Novus (G20717), and BoNT/A monoclonal: abcam (ab40786). Unfortunately, we were not able to obtain satisfactory staining with any of these reagents.

### Imaging and image analysis

High-throughput imaging was carried out using an automation-enabled Opera Phenix system with a 20 X objective (Revvity). GFP, RFP, and Hoechst channels were imaged and image analysis was done using Harmony and Columbus software (Revvity). In brief, nuclei counts were generated using the Hoechst channel while cell masks were obtained using the RFP channel. The nucleus region was excluded from the RFP cell mask to derive the cytoplasm region. The GFP Mean Fluorescent Intensity (MFI) was measured in the cytoplasm region and the ratio of GFP: RFP MFIs was calculated using Formula Output (A/B).

### Data formatting and normalization

Genome-wide RNAi screen data was imported from Columbus software and analyzed in ScreenSifter software as previously described (*Kumar et al., 2013*). GFP/RFP ratio was normalized to controls and Z-score graphs were plotted (*Figure 3—source data 1*).

### Statistical analysis
#### Western blot analysis

Rencell VM cells were transfected with siRNAs in a 10 cm dish for 3 days. On the third day, Cells were washed twice using ice-cold PBS before scraping in PBS. Cells were centrifuged at 300 g for 5 min at 4 °C and were lysed with ice-cold lysis buffer (50 mM Tris [pH 8.0, 4 °C], 200 mM NaCl), 0.5% NP-40 alternative, 1 mM DTT, and Complete Protease Inhibitor (Roche) for 30 min with gradual agitation before clarification of samples by centrifugation at 10,000 g for 10 min at 4 °C. Samples were diluted in lysis buffer with 2 X SDS loading buffer and boiled at 95 °C for 2 min. They were then resolved by SDS-PAGE electrophoresis using bis-tris NuPage gels as per manufacturer's instructions (Invitrogen) and transferred to PVDF membranes which was blocked using 3% BSA dissolved in TBST (50 mM Tris [pH8.0, 4 °C], 150 mM NaCl, and 0.1% Tween 20) at room temperature for 1 hr before incubation with antibodies as manufacturer's instructions.

## Acknowledgements

This study was funded by a grant from IPSEN and by an Industry Alignment Fund Grant from A*STAR. Jacquie Maignel, Laurent Pons, and Matthew Beard are employees of Ipsen. Jeremy Yeo is a former employee of IMCB, A*STAR. Keith Foster and Omar Loss are former employees of Ipsen. F Bard is

currently funded by a grant 'Leader in Oncology' from the Fondation 'ARC pour la recherche sur le cancer' and by a Chaire d'Excellence from AMIDEX: AMX-20-CE-03.

## Additional information

### Competing interests

Omar Loss, Keith Foster: former employee of Ipsen. Jacquie Maignel, Laurent Pons, Matthew Beard: employee of Ipsen. Frederic Bard: Reviewing editor, eLife. The other authors declare that no competing interests exist.

### Funding

| Funder | Grant reference number | Author |
|---|---|---|
| Institute of Molecular and Cell Biology | | Jeremy C Yeo<br>Felicia P Tay<br>Frederic Bard |
| Aix-Marseille Université | AMX-20-CE-03 | Frederic Bard |
| Fondation ARC pour la Recherche sur le Cancer | Leader Oncology | Frederic Bard<br>Rebecca Bennion |
| Ipsen | | Omar Loss<br>Jacquie Maignel<br>Laurent Pons<br>Keith Foster<br>Matthew Beard<br>Frederic Bard |

The funders had no role in study design, data collection and interpretation, or the decision to submit the work for publication.

### Author contributions

Jeremy C Yeo, Conceptualization, Investigation, Methodology; Felicia P Tay, Investigation, Methodology; Rebecca Bennion, Data curation, Investigation, Visualization, Writing – review and editing; Omar Loss, Resources, Validation; Jacquie Maignel, Conceptualization, Resources, Supervision; Laurent Pons, Resources, Formal analysis, Supervision; Keith Foster, Conceptualization, Resources, Project administration; Matthew Beard, Conceptualization, Resources, Formal analysis, Project administration; Frederic Bard, Conceptualization, Data curation, Formal analysis, Supervision, Writing – original draft, Writing – review and editing

### Author ORCIDs

Jeremy C Yeo http://orcid.org/0009-0009-6218-024X
Matthew Beard http://orcid.org/0000-0003-3249-7851
Frederic Bard https://orcid.org/0000-0002-3783-4805

Reviewer #1 (Public review): https://doi.org/10.7554/eLife.92806.3.sa1
Reviewer #2 (Public review): https://doi.org/10.7554/eLife.92806.3.sa2
Reviewer #3 (Public review): https://doi.org/10.7554/eLife.92806.3.sa3
Author response https://doi.org/10.7554/eLife.92806.3.sa4

## Additional files

### Supplementary files

• Supplementary file 1. Gene lists and gene ontology analysis for SV2 regulators. The file is separated into SV2 positive (+) and SV2 negative (-) regulators. Gene ontology analysis is included for Molecular Function, Biological Process, Cellular Component, and KEGG pathways.

• Supplementary file 2. Gene lists of BoNTA regulators (+) and BoNTA(-) regulators. The file includes the genes and the gene annotations.

• MDAR checklist

## Data availability

All data generated or analyzed supporting this study are included in the manuscript and supporting files.

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
