## [Editor Report · eLife assessment]

In this **valuable** manuscript, Yeo et al. describe new methods for assessing the intracellular itinerary of Botulinum neurotoxin A (BoNT/A), a potent toxin used in clinical and cosmetic applications. The current manuscript challenges previously held views on how the catalytic portion of the toxin makes its way from the endocytic compartment to the cytosol, to meet its substrates. The approach taken is deemed innovative and the experiments are carefully performed, presenting **solid** evidence for some of the drawn conclusion; however, the conclusions one may draw from the experimental results are somewhat limited, as it is possible that the scope of their findings could be restricted to the specific neuron model and molecular tools that were used. This paper could be of interest to both cell biologists and physicians.

---

## [Referee Report · Reviewer #1 (Public review)]

As outlined in my previous public review, Yeo et al. revised the current neuronal intoxication model, common to all serotypes of botulinum neurotoxins. Using a combination of genetic and imaging approaches, they demonstrate that upon internalization, BoNT/A-containing endosomes undergo retro-axonally trafficking to the neuronal soma. Within the soma, this particular serotype then traffics to the endoplasmic reticulum (ER) via the Golgi apparatus. At the ER, the SEC61 translocon complex facilitates the translocation of BoNT/A's metalloprotease domain (light chain, LC) from the ER lumen into the cytosol, where the thioredoxin reductase/thioredoxin system and HSP complexes release and refold the catalytic LC. Subsequently, the LC diffuses and cleaves SNAP25 first in the soma before reaching neurites and synapses.

Although I still acknowledge the well-executed and thoroughly analyzed genome-wide RNAi screen, I must once again highlight significant pitfalls and weaknesses in the paper due to the lack of essential controls and validations. Consequently, I suggest readers to approach the authors' findings with caution, as they may be limited to the combination of one specific cellular model and genetic engineering tools. During the revision process, authors declined to conduct additional experiments that could have strengthened their main conclusions. These include, but are not limited to:

(1) Investigating weather in the newly generated cell line Red-SNAPR, the GFP fragment produced upon toxin cleavage degrades more rapidly in the soma compared to axon terminals, possibly due to differences in proteasome activity in these two compartments.

(2) Validating toxin cleavage activity in the soma before reaching synapses by conducting an additional and more physiological approach, a time course experiment using native BoNT/A and staining BoNT/A-cleaved SNAP25 with specific antibodies.

(3) Assessing whether the addition of mNG1-11 to the LC affects the translocation process itself and quantifying the mean fluorescence intensity (MFI) per cell, taking into consideration the amount of HA-tagged Cyt-mG1-10, which appears predominantly expressed in the cytosol and less detected in neurites. This raises the question of potential bias toward the cell soma in this assay.

(4) Validating major hits (e.g., VPS34 and Sec61) by performing WB or IF analysis to test the cleavage of endogenous SNAP25.

Additionally, during the revision process, the authors raised concerns about the level of scrutiny applied by this reviewer, particularly in comparison to the seminal study of Lilia K. Koriazova & Mauricio Montal published in Nature Structural Biology (PMID: 12459720). In this 2003 paper, Montal's lab pioneered the use of single-channel recordings and substrate proteolysis analysis to reconstitute the translocation of BoNT/A light chain protease across an artificial lipid bilayer via the channel formed by its heavy chain. The authors highlighted that, when converting the experimental conditions from the aforementioned paper into molarity, it appears that the cis compartment was loaded with 10−8 M BoNT/A, and the reported translocated protease activity (measured by substrate cleavage) is equivalent to 10−17 M. This implies that only about 1 LC molecule in 100 million has crossed the membrane. The calculation performed by authors is indeed accurate. However, readers should be informed about another piece of information present in the same paper that might help them to clarify this important point. Koriazova & Montal, by discussing this experiment, have pointed out that this value (10−17 M) corresponds to ≈3600 LC molecules, a number closed to the maximum number of channels that can be formed under the used experimental conditions. Indeed, from the same paper, quotation: 'This number is in close agreement with the maximum number of channels inserted in the bilayer under the assay condition, ≈2000 (Fig. 3a), as estimated from macroscopic membrane conductance ∼1 × 105 pS and γ = 50 pS measured in 0.1 M KCl'. Another aspect that Yeo et al. forgot to mention in their rebuttal letter is that the system used by Koriazova & Montal lacks any chaperones in the trans compartment. Nowadays, we know that upon translocation, the refolding of the L chain is aided by Hsp90 (Azarnia Tehran et al., Cellular microbiology, 2017). Keeping this in mind, is not unrealistic to hypothesize that the number of LC molecules calculated more than 22 years ago by Koriazova & Montal (in an indirect way by checking SNAP25 cleavage using an ELISA-based assay) might be an underestimation. Indeed, the addition of Hsp90 in their system might aid in the refolding of LC molecules that, even if they have successfully be translocated, might not cleave the substrate due to their unfolded state.

As active scientist, I understand the challenges of peer review and publication, which can often be slow and frustrating involving seemingly endless rounds of review. Therefore, I am in favor of the new eLife publishing model. Indeed, this paper has already been published as Reviewed Preprints and will soon be declared as the final Version of Record, accompanied by this public review. Having said that, I hope that the readers of this journal and future scientists will prove me wrong. I hope they will engage with this paper, providing comments, validations (which are currently missing), and citations as frequently as they did for the seminal works of Koriazova & Montal.

---

## [Referee Report · Reviewer #2 (Public review)]

Summary:

The study by Yeo and co-authors addresses a long-lasting issue about botulinum neurotoxin (BoNT) intoxication. The current view is that the toxin binds to its receptors at the axon terminus by its HCc domain and is internalized in recycled neuromediator vesicles just after release of the neuromediators. Then, the HCn domain assists the translocation of the catalytic light chain (LC) of the toxin through the membrane of these endocytic vesicles into the cytosol of the axon terminus. There, the LC cleaves its SNARE substrate and blocks neurosecretion. However, other views involving kinetic aspects of intoxication suggest that the toxin follows the retrograde axonal transport up to the nerve cell body and then back to the nerve terminus before cleaving its substrate.

In the current study, the authors claim that the BoNT/A (isotype A of BoNT) not only progresses to the cell body but once there, follows the retrograde transport trafficking pathway in a retromer-dependent fashion, through the Golgi apparatus, until reaching the endoplasmic reticulum. Next, the LC dissociates from the HC (a process not studied here) and uses the translocon Sec61 machinery to retro-translocate into the cytosol. Only then, the LC traffics back to the nerve terminus following the anterograde axonal transport. Once there, LC cleaves its SNARE substrate (SNAP25 in the case of BoTN/A) and blocks neurosecretion.

To reach their conclusion, Yeo and co-authors use a combination of engineered tools: a cell line able to differentiate into neurons (ReNcell VN), a reporter dual fluorescent protein derived from SNAP25, the substrate of BoNT/A (called SNAPR), the use of either native BoNT/A or a toxin to which three fragment 11 of the reporter fluorescent protein Neon Green (mNG) are fused to the N-terminus of the LC (BoNT/A-mNG11x3), and finally ReNcell VN transfected with mNG1-10 (a protein consisting of the first 10 beta strands of the mNG).

SNAPR is stably expressed all over in the ReNcell VN. SNAPR is yellow (red and green) when intact and becomes red only when cleaved by BoNT/A LC, the green tip being degraded by the cell. When the LC of BoNT/A-mNG11x3 reaches the cytosol in ReNcell VN transfected by mNG1-10, the complete mNG is reconstituted and emits a green fluorescence.

In the first experiment, the authors show that the catalytic activity of the LC appears first in the cell body of neurons where SNAPR is cleaved first. This phenomenon starts 24 h after intoxication and progresses along the axon towards the nerve terminus during an additional 24 h. In a second experiment, the authors intoxicate the ReNcell VN transfected by mNG1-10 using the BoNT/A-mNG11x3. The fluorescence appears also first in the soma of neurons, then diffuses in the neurites in 48 h. The conclusion of these two experiments is that translocation occurs first in the cell body and that the LC diffuses in the cytosol of the axon in an anterograde fashion.

In the second part of the study, the authors perform a siRNA screen to identify regulators of BoNT/A intoxication. Their aim is to identify genes involved in intracellular trafficking of the toxin and translocation of the LC. Interestingly, they found positive and negative regulators of intoxication. Regulators could be regrouped according to the sequential events of intoxication. Genes affecting binding to the cell-surface receptor (SV2) and internalization. Genes involved in intracellular trafficking. Genes involved in translocation such as reduction of the disulfide bond linking the LC to the HC and refolding in the cytosol. Genes involved in signaling such as tyrosine kinases and phosphatases. All these groups of genes may be consistent with the current view of BoNT intoxication within the nerve terminus. However, two sets of genes were particularly significant to reach the main conclusion of the work and definitely constitute an original finding important to the field. One set of genes consists in those of the retromer, the other relates to the Sec61 translocon. This should indicate that once endocytosed, the BoNT traffics from the endosomes to Golgi apparatus, then to the ER. Ultimately, the LC should translocate from the ER lumen to the cytosol using the Sec61 translocon. The authors further control that the SV2 receptor for the BoNT/A traffics along the axon in a retromer-dependent fashion and that BoNT/A-mNG11x3 traverses the Golgi apparatus by fusing the mNG1-10 to a Golgi resident protein.

Strengths:

The findings in this work are convincing. The experiments are carefully done and are properly controlled. In the first part of the study, both the activity of the LC is monitored together with the physical presence of the toxin. In the second part of the work, the most relevant genes that came out of the siRNA screen are checked individually in the ReNcell VN / BoNT/A reporter system to confirm their role in BoNT/A trafficking and retro-translocation.

These findings are important to the fields of toxinology and medical treatment of neuromuscular diseases by BoNTs. They may explain some aspects of intoxication such as slow symptom onset, aggravation and appearance of central effects.

Weaknesses:

The findings antagonize the current view of the intoxication pathway that is sustained by a vast amount of observations. The findings are certainly valid, but their generalization as the sole mechanism of BoNT intoxication should be tempered. These observations are restricted to one particular neuronal model and engineered protein tools. Other models such as isolated nerve/muscle preparations display nerve terminus paralysis within minutes rather than days. Also, the tetanus neurotoxin (TeNT), which mechanism of action involving axonal transport to the posterior ganglia in the spinal cord is well described, takes between 5 and 15 days. It is thus possible that different intoxication mechanisms co-exist for BoNTs or even vary depending on the type of neurons.

Although the siRNA experiments are convincing, it would be nice to reach the same observations with drugs affecting the endocytic to Golgi to ER transport (such as Retro-2, golgicide or brefeldin A) and the Sec61 retrotranslocation (such as mycolactone). Then, it would be nice to check other neuronal systems for the same observations.

---

## [Referee Report · Reviewer #3 (Public review)]

Summary:

The manuscript by Yeo et al. investigates the intracellular trafficking of Botulinum neurotoxin A (BoNT/A), a potent toxin used in clinical and cosmetic applications. Contrary to the prevailing understanding of BoNT/A translocation into the cytosol, the study suggests a retrograde migration from the synapse to the soma-localized Golgi in neurons. Using a genome-wide siRNA screen in genetically engineered neurons, the researchers identify over three hundred genes involved in this process. The study employs organelle-specific split-mNG complementation, revealing that BoNT/A traffics through the Golgi in a retromer-dependent manner before moving to the endoplasmic reticulum (ER). The Sec61 complex is implicated in the retro-translocation of BoNT/A from the ER to the cytosol. Overall, the research challenges the conventional model of BoNT/A translocation, uncovering a complex route from synapse to cytosol for efficient intoxication. The findings are based on a comprehensive approach, including the introduction of a fluorescent reporter for BoNT/A catalytic activity and genetic manipulations in neuronal cell lines. The conclusions highlight the importance of retrograde trafficking and the involvement of specific genes and cellular processes in BoNT/A intoxication.

Strengths:

The major part of the experiments are convincing. They are well-controlled and the interpretation of their results is balanced and sensitive.

Weaknesses:

To my opinion, the main weakness of the paper is that all experiments are performed using a single cellular system (RenVM neurons), as stated in the title. It is therefore unclear at the moment to what extent the findings in this paper can be generalized to other neuronal cell models / in vivo situation.

---

## [Author Response]

The following is the authors’ response to the original reviews.

**Public Reviews:**

**Reviewer #1 (Public Review):**
Summary:During the last decades, extensive studies (mostly neglected by the authors), using in vitro and in vivo models, have elucidated the five-step mechanism of intoxication of botulinum neurotoxins (BoNTs). The binding domain (H chain) of all serotypes of BoNTs binds polysialogangliosides and the luminal domain of a synaptic vesicle protein (which varies among serotypes). When bound to the synaptic membrane of neurons, BoNTs are rapidly internalized by synaptic vesicles (SVs) via endocytosis. Subsequently, the catalytic domain (L chain) translocates, a process triggered by the acidification of these organelles. Following translocation, the disulfide bridge connecting the H chain with the L chain is reduced by the thioredoxin reductase/thioredoxin system, and it is refolded by the chaperone Hsp90 on SV's surface. Once released into the cytosol, the L chains of different serotypes cleave distinct peptide bonds of specific SNARE proteins, thereby disrupting neurotransmission.In this study, Yeo et al. extensively revise the neuronal intoxication model, suggesting that BoNT/A follows a more complex intracellular route than previously thought. The authors propose that upon internalization, BoNT/A-containing endosomes are retro-axonally trafficked to the soma. At the level of the neuronal soma, this serotype then traffics to the endoplasmic reticulum (ER) via the Golgi apparatus. The ER SEC61 translocon complex facilitates the translocation of BoNT/A's LC from the ER lumen into the cytosol, where the thioredoxin reductase/thioredoxin system and HSP complexes release and refold the catalytic L chain. Subsequently, the L chain diffuses and cleaves SNAP25 first in the soma before reaching neurites and synapses.Strengths:I appreciate the authors' efforts to confirm that the newly established methods somehow recapitulate aspects of the BoNTs mechanism of action, such as toxin binding and uptake occurring at the level of active synapses. Furthermore, even though I consider the SNAPR approach inadequate, the genome-wide RNAi screen has been well executed and thoroughly analyzed. It includes well-established positive and negative controls, making it a comprehensive resource not only for scientists working in the field of botulinum neurotoxins but also for cell biologists studying endocytosis more broadly.Weaknesses:I have several concerns about the authors' main conclusions, primarily due to the lack of essential controls and validation for the newly developed methods used to assess toxin cleavage and trafficking into neurons. Furthermore, there is a significant discrepancy between the proposed intoxication model and existing studies conducted in more physiological settings. In my opinion, the authors have omitted over 20 years of work done in several labs worldwide (Montecucco, Montal, Schiavo, Rummel, Binz, etc.). I want to emphasize that I support changes in biological dogma only when these changes are supported by compelling experimental evidence, which I could not find in the present manuscript.

We thank the reviewer for his reading and comments and for pointing out the discrepancy between our proposed model and the existing model. However, we respectfully disagree with the phrase of “extensive studies have elucidated the five-steps mechanism of intoxication…”. This sentence and the following imply that the model is well-established and demonstrated. It also highlights how the reviewer is convinced about this previous model.

We contest this model for theoretical reasons and contest the strength of evidences that support it. We previously included references to previous work showing that the model is also being challenged by others. In light of the reviewer’s comments, we incluced more references in the introduction and we also explicit our main theoretical concern in the introduction:

“Arguably, the main problem of the model is its failure to propose a thermodynamically consistent explanation for the directional translocation of a polypeptidic chain across a biologial membrane. Other known instances of polypeptide membrane translocation such as the co-translational translocation into the ER indicate that it is an unfavorable process, which consumes significant energy (Alder and Theg 2003). ”

We also added the following text in the Discussion to address with the reviewer’s concerns: “Our study contradicts the long-established model of BoNT intoxication, which is described in several reviews specifically dedicated to the subject 1–4. In short, these reviews support the notion that BoNT are molecular machines able to mediate their own translocation across membranes; this notion has convinced some cell biologists interested in toxins and retrograde traffic, who describe BoNT mode of translocation in their reviews 5,6.

But is this notion well supported by data? A careful examination of the primary literature reveals that early studies indeed report that BonTs form ion channels at low pH values 7,8. These studies have been extended by the use of patch-clamp 9,10. These works and others lead to various suppositions on how the toxin forms a channel and translocate the LC 1,11 .

However, only a single study claims to reconstitute in vitro the translocation of BonT LC across membranes 12. In this paper, the authors report using a system of artificial membranes separating two aqueous compartments. They load the toxin in the cis compartment and measure the protease activity in the trans compartment after incubation. However, when the experimental conditions described are actually converted in terms of molarity, it appears that the cis compartment was loaded at 10e-8M BonT and that the reported translocated protease activity is equivalent to 10e-17 M (Figure 3D, 12). Thus, in this experiment, about 1 LC molecule in 100 millions has crossed the membrane. Such extremely low transfert rate does not tally with the extreme efficiency of intoxication in vivo, even while taking into account the difference between artificial and biological membranes.

In sum, a careful analysis of the primary literature indicate that while there is ample evidence that BoNTs have the ability to affect membranes and possibly create ion channels, there is actually no credible evidence that these channels mediate translocation of the LC. As mentioned earlier, it is not clear how such a self-translocation mechanism would function thermodynamically. By contrast, our model proposes a mechanism without a thermodynamic problem, is consistent with current knowledge about other protein toxins, such as PE, Shiga and Ricin, and can help explain previously puzzling features of BonT effects. It is worth noting that a similar self-translocation model was proposed for other protein toxins such as Pseudomonas exotoxin, which have similar molecular organisation as BonT (68). However, it has since been demonstrated that the PE toxins require cellular machinery, in particular in the ER, for intoxication (21,69,70).”

**Reviewer #2 (Public Review):**
Summary:The study by Yeo and co-authors addresses a long-lasting issue about botulinum neurotoxin (BoNT) intoxication. The current view is that the toxin binds to its receptors at the axon terminus by its HCc domain and is internalized in recycled neuromediator vesicles just after the release of the neuromediators. Then, the HCn domain assists the translocation of the catalytic light chain (LC) of the toxin through the membrane of these endocytic vesicles into the cytosol of the axon terminus. There, the LC cleaves its SNARE substrate and blocks neurosecretion. However, other views involving kinetic aspects of intoxication suggest that the toxin follows the retrograde axonal transport up to the nerve cell body and then back to the nerve terminus before cleaving its substrate.In the current study, the authors claim that the BoNT/A (isotype A of BoNT) not only progresses to the cell body but once there, follows the retrograde transport trafficking pathway in a retromer-dependent fashion, through the Golgi apparatus, until reaching the endoplasmic reticulum. Next, the LC dissociates from the HC (a process not studied here) and uses the translocon Sec61 machinery to retro-translocate into the cytosol. Only then, does the LC traffic back to the nerve terminus following the anterograde axonal transport. Once there, LC cleaves its SNARE substrate (SNAP25 in the case of BoTN/A) and blocks neurosecretion.To reach their conclusion, Yeo and co-authors use a combination of engineered tools: a cell line able to differentiate into neurons (ReNcell VN), a reporter dual fluorescent protein derived from SNAP25, the substrate of BoNT/A (called SNAPR), the use of either native BoNT/A or a toxin to which three fragment 11 of the reporter fluorescent protein Neon Green (mNG) are fused to the N-terminus of the LC (BoNT/A-mNG11x3), and finally ReNcell VN transfected with mNG1-10 (a protein consisting of the first 10 beta strands of the mNG).SNAPR is stably expressed all over in the ReNcell VN. SNAPR is yellow (red and green) when intact and becomes red only when cleaved by BoNT/A LC, the green tip being degraded by the cell. When the LC of BoNT/A-mNG11x3 reaches the cytosol in ReNcell VN transfected by mNG1-10, the complete mNG is reconstituted and emits a green fluorescence.In the first experiment, the authors show that the catalytic activity of the LC appears first in the cell body of neurons where SNAPR is cleaved first. This phenomenon starts 24 hours after intoxication and progresses along the axon towards the nerve terminus during an additional 24 hours. In a second experiment, the authors intoxicate the ReNcell VN transfected by mNG1-10 using the BoNT/A-mNG11x3. The fluorescence appears also first in the soma of neurons, then diffuses in the neurites in 48 hours. The conclusion of these two experiments is that translocation occurs first in the cell body and that the LC diffuses in the cytosol of the axon in an anterograde fashion.In the second part of the study, the authors perform a siRNA screen to identify regulators of BoNT/A intoxication. Their aim is to identify genes involved in intracellular trafficking of the toxin and translocation of the LC. Interestingly, they found positive and negative regulators of intoxication. Regulators could be regrouped according to the sequential events of intoxication.Genes affecting binding to the cell-surface receptor (SV2) and internalization. Genes involved in intracellular trafficking. Genes involved in translocation such as reduction of the disulfide bond linking the LC to the HC and refolding in the cytosol. Genes involved in signaling such as tyrosine kinases and phosphatases. All these groups of genes may be consistent with the current view of BoNT intoxication within the nerve terminus. However, two sets of genes were particularly significant to reach the main conclusion of the work and definitely constitute an original finding important to the field. One set of genes consists of those of the retromer, and the other relates to the Sec61 translocon. This should indicate that once endocytosed, the BoNT traffics from the endosomes to the Golgi apparatus, and then to the ER. Ultimately, the LC should translocate from the ER lumen to the cytosol using the Sec61 translocon. The authors further control that the SV2 receptor for the BoNT/A traffics along the axon in a retromer-dependent fashion and that BoNT/A-mNG11x3 traverses the Golgi apparatus by fusing the mNG1-10 to a Golgi resident protein.Strengths:The findings in this work are convincing. The experiments are carefully done and are properly controlled. In the first part of the study, both the activity of the LC is monitored together with the physical presence of the toxin. In the second part of the work, the most relevant genes that came out of the siRNA screen are checked individually in the ReNcell VN / BoNT/A reporter system to confirm their role in BoNT/A trafficking and retro-translocation.These findings are important to the fields of toxinology and medical treatment of neuromuscular diseases by BoNTs. They may explain some aspects of intoxication such as slow symptom onset, aggravation, and appearance of central effects.Weaknesses:The findings antagonize the current view of the intoxication pathway that is sustained by a vast amount of observations. The findings are certainly valid, but their generalization as the sole mechanism of BoNT intoxication should be tempered. These observations are restricted to one particular neuronal model and engineered protein tools. Other models such as isolated nerve/muscle preparations display nerve terminus paralysis within minutes rather than days. Also, the tetanus neurotoxin (TeNT), whose mechanism of action involving axonal transport to the posterior ganglia in the spinal cord is well described, takes between 5 and 15 days. It is thus possible that different intoxication mechanisms co-exist for BoNTs or even vary depending on the type of neurons.Although the siRNA experiments are convincing, it would be nice to reach the same observations with drugs affecting the endocytic to Golgi to ER transport (such as Retro-2, golgicide or brefeldin A) and the Sec61 retrotranslocation (such as mycolactone). Then, it would be nice to check other neuronal systems for the same observations.

We thank the reviewer for the careful reading and comments of our manuscript. The reference to “a vast amount of observation” is a similar argument to the Reviewer 1 and used to suggest that our study may not be applicable as a general mechanism.

We respectfully disagree as described above and posit on the contrary that the model we propose is much more likely to be general than the model presented in current reviews for the several reasons cited (see added text in Introduction and Discussion). While we agree that more work is needed to confirm the proposed mechanisms of BonT translocation in other models, these experiments fall outside the perimeter of our study.

The fact that nerve/muscle preparations of BonT activity have relatively fast kinetics does not pose a contradiction to our model. Our model reveals primarily the requirement for trafficking to the ER membranes. This ER targeting requires trafficking through the Golgi complex, in turn explaining the requirement for trafficking to the soma of neurons in the experimental system we used. However, in neuronal cells in vivo, Golgi bodies can be found along the lenght of the axon, thus BonT may not always require trafficking to the soma of the affected cells. The time required for intoxication could thus vary greatly depending on the neuronal structural organisation.

TenT is proposed to transfer from excitatory neurons into inhibitory neurons before exerting its action. While the detailed mechanism of this fascinating mechanism remain to be explored, it clearly falls beyond the purview of this manuscript.

Regarding the use of drugs, we agree that it would be a nice addition; unfortunately we are unable to perform such experiments at this stage. Setting up a large scale siRNA screen for BonT mechanism of action is challenging as it requires a special facility with controlled access and police authorisation (in Singapore) given the high toxicity of this molecule. Unfortunately, the authorisations have now lapsed.

**Reviewer #3 (Public Review):**Summary:The manuscript by Yao et al. investigates the intracellular trafficking of Botulinum neurotoxin A (BoNT/A), a potent toxin used in clinical and cosmetic applications. Contrary to the prevailing understanding of BoNT/A translocation into the cytosol, the study suggests a retrograde migration from the synapse to the soma-localized Golgi in neurons. Using a genome-wide siRNA screen in genetically engineered neurons, the researchers identified over three hundred genes involved in this process. The study employs organelle-specific split-mNG complementation, revealing that BoNT/A traffics through the Golgi in a retromer-dependent manner before moving to the endoplasmic reticulum (ER). The Sec61 complex is implicated in the retro-translocation of BoNT/A from the ER to the cytosol. Overall, the research challenges the conventional model of BoNT/A translocation, uncovering a complex route from synapse to cytosol for efficient intoxication. The findings are based on a comprehensive approach, including the introduction of a fluorescent reporter for BoNT/A catalytic activity and genetic manipulations in neuronal cell lines. The conclusions highlight the importance of retrograde trafficking and the involvement of specific genes and cellular processes in BoNT/A intoxication.Strengths:The major part of the experiments are convincing. They are well-controlled and the interpretation of their results is balanced and sensitive.Weaknesses:To my opinion, the main weakness of the paper is in the interpretation of the data equating loss of tGFP signal (when using the Red SNAPR assay) with proteolytic cleavage by the toxin. Indeed, the first step for loss of tGFP signal by degradation of the cleaved part is the actual cleavage.However, this needs to be degraded (by the proteasome, I presume), a process that could in principle be affected (in speed or extent) by the toxin.

We thank the reviewer for his comments and careful reading of our manuscript.

Regarding the read-out of the assay, we agree that the assay could be sensitive to alteration in the protein degradation pathway. We have added the following sentence in the Discussion to take it into account:

“As noted by one reviewer, the assay may be sensitive to perturbation in the general rate of protein degradation, a consideration to keep in mind when evaluating the results of large scale screens.”

While this may be valid for some hits in the general list, it is important to note that the main hits have been shown to affect toxin trafficking by an independent, orthogonal assay based on the split GFP reconstitution.

**Recommendations to authors:**

**Reviewer #1 (Recommendations For The Authors):**
(1) To assess the activity of BoNT/A in neurons, Yeo et al. have generated a neuronal stem line referred to as SNAPR. This cell line stably expresses a chimeric reporter protein that consists of SNAP25 flanked at its N-terminus with a tagRFPT and at its C-terminus with a tagGFP. After exposure to BoNT/A, SNAP25 is cleaved and, the C-terminal tGFP-containing moiety is rapidly degraded. I have many doubts about the validity of the described method. Indeed, BoNT/A activity is analysed in an indirect way by quantifying the degradation of the GFP moiety generated after toxin cleavage (Fig. 2). In this regard, the authors should consider that their approach is dependent, not only on the toxin's metalloprotease activity but also on the functionality of the proteasome in neurons. Therefore, considering the current dataset, it is impossible to rule out the possibility that the progression of GFP signal loss from the soma to the neurite terminals may be attributed to the different proteasome activity in these compartments. Is it conceivable that the GFP fragment generated upon toxin cleavage degrades more rapidly in the soma in comparison to axonal terminals? This alternative explanation could challenge the conclusion drawn in Fig. 2.

The reviewer’s alternative explanation disregards the experiments performed with the split-GFP complementation approach, which indicate translocation in the soma first. The split GFP reporter is not dependent on the proteasome activity. It also disregard the genetic data implicating many genes involved in membrane retrograde traffic, which are also not consistent with the hypothesis of the reviewer. These genes depletions not only affect SNAPR degradation but also BoNT/A-mNG11 trafficking: thus, their effect cannot be attributed to an completely hypothetical spatial heterogeneous distribution of the proteasome.

For this reason, I strongly suggest using a more physiological approach that does not depend on proteasomal degradation or on the expression of the sensor in neurons. The authors should consider performing a time course experiment following intoxication and staining BoNT/A-cleaved SNAP25 by using specific antibodies (see Antonucci F. et al., Journal of Neuroscience, 2008 or Rheaume C. et al., Toxins 2015).

For the above reason, we do not agree with the pressing importance of confirming by a third method using specific antibodies; especially considering that BonT is very difficult to detect in cells when incubated at physiological levels. By the way, the cited paper, by Antonucci F; et al. documents long distance retrograde traffic of BonT/A, which is in line with our data.

An alternative approach could involve the use of microfluidic devices that physically separate axons from cell bodies. Such a separation will allow us to test the authors' primary conclusion that SNAP25 is initially cleaved in the soma. The suggested experiments will also rule out potential overexpression artifacts that could influence the authors' conclusions when using the newly developed SNAPR approach. Without these additional experiments, the authors' main conclusion that SNAP25 is cleaved first in the neuronal soma rather than at the nerve terminal is inadequate.

As discussed above we disagree about the doubts raised by the reviewer: we present three types of evidences (SNAPR, split GFP and genetic hits) and they all point in the same direction. Thus, we respectfully doubt that a fourth approach would convince this reviewer. To note, we have attempted to use microfluidics devices as suggested by the reviewer, however, the Ren-VM neurons were not able to extend axons long enough across the device.

(2) To detect BoNT/A translocation into the cytosol, the authors have used a complementation assay by intoxicating ReNcell VM cell expressing a cytosolic HA-tagged split monomeric NeonGreen (Cyt-mNG1-10) with an engineered BoNT/A, where the catalytic domain (LC) was fused to mNG1-11. When drawing conclusions regarding the detection of cytosolic LC in the neuronal soma, the authors should highlight the limitations of this assay and explicitly describe them to the readers. Firstly, the authors need to investigate whether the addition of mNG1-11 to the LC affects the translocation process itself (by comparing with a WT, not tagged, LC).Additionally, from the data shown in Fig. 2C, it is evident that the Cyt-mNG1-10 is predominantly expressed in the cytosol and less detected in neurites. This raises the question of whether there might be a bias for the cell soma in this assay. To address this important concern, I suggest quantifying MFI per cell (Fig. 2D) taking into consideration the amount of HA-tagged Cyt-mNG1-10. Furthermore, I strongly suggest targeting mNG1-10 to synapses and performing a similar time course experiment to observe when LC translocation occurs at nerve terminals. Alternative experiments, to prove that BoNT/A requires retrograde trafficking before it can translocate, may be done to repeat the experiments shown in Fig. 2D in the presence of inhibitors (or by KD some of the hits identified as microtubule stabilizers) that should interfere with BoNT/A trafficking to the neuronal somata. Without these additional experiments, the authors' main conclusion that the BoNT/A catalytic domain is first detected in the neuronal soma rather than at the nerve terminal is very preliminary.

Similarly as for the SNAPR assay, the reviewer is raising the level of doubt to very high levels. We respect his thoroughness and eagerness to question the new model. However, we note that a similar level of scrutiny does not apply to the prevalent competitive model. Indeed, the data supporting the self-translocation model is based on a single in vitro experiment published in one panel as we have explain din the discussion (see above).

(3) In the genome-wide RNAi screening, rather than solely assessing SV2 surface levels, it would have been beneficial to directly investigate BoNT/A binding to the neuronal membrane. For instance, this could have been achieved by using a GFP-tagged HC domain of BoNT/A. At present, the authors cannot exclude the possibility that among the 135 hits that did not affect SV2 levels, some might still inhibit BoNT/A binding to the neuronal surface. These concerns, already exemplified by B4CALT4 (which is known to be involved in the synthesis of GT1b), should be explicitly addressed in the main text.

We agree with the reviewer that perturbation of binding of BonT is possible. We added the following text:

“Network analysis reveals regulators of signaling, membrane trafficking and thioreductase redox state involved in BoNT/A intoxication

Among the positive regulators of the screen, 135 hits did not influence significantly surface SV2 levels and are thus likely to function in post-endocytic processes (Supplementary Table 2). However, we cannot formerly exclude that they could affect binding of BonT to the cell surface independently of SV2.”

(4) The authors should clearly state which reagents they have tried to use in order to explain the challenges they faced when directly testing the trafficking of BoNT/A. The accumulation of Dendra-SV2 bulbous structures at the neurite tips in VPS35-depleted cells could be interpreted as a sign of neuronal stress/death. Have the authors investigated other proteins that do not undergo retro-axonal trafficking in a retromer-dependent manner? This control is essential. In this regard, the use of a GFP-tagged HC domain of BoNT/A could prove to be quite helpful.

We tried multiple commercially available antibodies against BonT but we could not get a very good signal. The postdoc in charge of this project has now gone to greener pastures and we are not in the capacity to provide the details corresponding to these antibodies. We di dnot observe significant cell death after VPS-35 knockdown at the time of the experiment, however longe rterm treatment might result in toxicity indeed.

(5) Considering my concerns related to the SNAPR system and the complementation assay to studySNAP25 cleavage and BoNT/A trafficking, I suggest validating some of their major hits (ex. VPS34 and Sec61) by performing WB or IF analysis to examine the cleavage of endogenous SNAP25. Furthermore, the authors should test VPS35 depletion in the context of the experiments performed in Fig. 6G-H, by validating that this protein is essential for BoNT/A retrograde trafficking.

The reviewer concerns are well noted but as discussed above, the two systems we used are completely orthogonal. Thus, for the reviewer’s concerns to be valid, it would have to be two completely independent artefacts giving rise to the same result. The alternative explanation is that BonT/A translocates in the soma. The Ockham razor principle dictates that the simplest explanation is the likeliest.

(6) The introduction and the discussion section of this paper completely disregard more than 20 years of research conducted by several labs worldwide (Montecucco, Montal, Schiavo, Rummel, Binz, etc). The authors should make an effort to contextualize their data within the framework of these studies and address the significant discrepancies between their proposed intoxication model and existing research that clearly demonstrates BoNTs translocating upon the endocytic retrieval of SVs at presynaptic sites. Nevertheless, even assuming that the model proposed by the authors is accurate, numerous questions emerge. One such question is: How can the authors explain the exceptional toxicity of botulinum neurotoxin in an ex vivo neuromuscular junction preparation devoid of neuronal cell bodies (see Cesare Montecucco and Andreas Rummel's seminal studies)?

Please see above in the answer to public reviews.

(7) Scale bars should be added to all representative pictures.

This has been done. Thank you for the thorough reading of our manuscript.

*Reviewer #2(Recommendations For The Authors):**(1) The title overstates the results. It may be indicated "in differenciated ReNcell VM".

Title changed to: “Botulinum toxin intoxication requires retrograde transport and membrane translocation at the ER in RenVM neurons”

(2) In the provided manuscript there are two Figure 2 and no Figure 3. This made the reading and understanding extremely difficult and should be corrected. As a result, the Figure legends do not fit the numbering. There are also discrepancies between some Figure panels (A, B, C, etc), the text, and the Legends. All this needs to be carefully checked.

We apologize for the confusion as the manuscript as followed multiple rounds of revisions. We have carefully verified labels and legends.

(3) The BoNT/A-mNG11x3 may introduce some bias that could be discussed. Would these additional peptides block LC translocation from synaptic vesicles in the nerve termini? In addition, the mNG peptides that are unfolded before complementation may direct LC towards Sec61. These aspects should be discussed.

The comment would be valid if BoNT/A-mNG11x3 was the only approach used in the paper, however the SNAPR reporter is used with native BonT and shows data consistent with the split GFP approach.

(4) In the Figure about SV2 (Fig 3 or 4): The authors did not locate SV2. The cells seem not to have the same differentiated phenotype as in Figure 1 and Figure 2/3A.

We apologized above for the mislabeling. It is not clear what is the question here.

(5) The authors should check whether BoNT/A wt cleaves the endogeneous SNAP25 by western blot for instance in the original ReNcell VN before SNAPR engineering. This should be compared with wt SNAP25 cleavage by the BoNT/A-LC-mNG.

It is likely that BoNT/A-LC-mNG11 should have similar activity as it is only adding a small peptide at the end of the LC. At any rate, it is not clear why this is so important since both molecules translocate in the cytosol, with the same kinetics and in the same subcellular locale.

(6) Perhaps I did not understand. How can the authors exclude that what is observed is the kinetic overproduction of the reporter substrate SNAPR?

The authors could use SLO toxin (PNAS 98, 3185-3190, 2001) to permeabilize the cells all along their body and axon to introduce BoNT/A or LC (wt) and observe synchronized SNAPR cleavage throughout the cells.

The concept mentioned here is not very clear to us. The reviewer is proposing that the SNAPR is produced much more efficiently at the tips of the neurites and thus its cleavage takes longer to be detected and is apparent first in the soma?? With all due respect, this is a strange hypothesis, at odds with what we know of protein dynamics in the neurons (i.e. most proteins are largely made in the soma and transported or diffuse into the neurites).

Again, the two orthogonal approaches: split GFP and SNAPR reporter use different constructs and methods, yet converge on similar results. Perhaps, the incredulity of the reviewer might be more productively directed at the current data “demonstrating” the translocation of LC in the synaptic button?

(7) The authors could also use an essay on neurotransmitter release monitoring by electrophysiology measurements to check the functional consequences of the kinetic diffusion of LC activity along the axon. Can the authors exclude that some toxin molecules translocate from the endocytic vesicles and block neurotransmission within minutes or a few hours?

It is well established that inhibition of neurotransmission does not occur within minutes in vivo and in vitro, but rather within hours or even days. This kinetic delay is experienced by many patients and is one of the key argument against the current model of self-translocation at the synaptic vesicle level.

Minor remarks

Thank you for pointing out all these.

(1) Please check typos. There are many. Check space before the parenthesis, between numbers and h (hours), reference style etc.

Thank you. We have reviewed the text and try to eliminate all these instances.

(2) Line 90: The C of HC should be capitalized.

Fixed

(3) Line 107: add space between "neuronsDonato".

Fixed

(4) Line 109: space "72 h".

Fixed

(5) Line 115: a word is missing ? ...to show retro-axonal... ? Please clarify this sentence.

Fixed

(6) Figure 1E: does nm refer to nM (nanomolar)? Please correct. No mention of panel F.

Fixed

(7) Line 161: do you mean ~16 µm/h? Please correct.

Fixed

(8) Line 168, words are missing.

Fixed, thank you

We verified that Cyt-mNG1-10 was expressed using the HA tag, the expression was homogeneously distributed in differentiated neurons and we observed no GFP signal (Figure2C).

(9) Line 171: Isn't mNG 11 the eleventh beta strand of the neon green fluorescent protein, not alpha helix? Otherwise, can the authors confirm it acquires the shape of an alpha helix? Same at line 326.

We have corrected the mistake; thanks for pointing it out.

(10) Figure 2 is doubled. The legend of Fig 2 refers to Figure 3. There is no legend for Figure 2. Then, some figures are shifted in their numbering.

Fixed

(11) The fluorescence in the cell body must appear before the fluorescence in the axon due to higher volume. Please discuss.

The fluorescence progresses in the neurites extensions in a centripetal fashion. The volume of the neurite near the cell body is not significantly different from the end of the neurite. Thus the fluorescence data is consistent with translocation in soma and not with an effect due to higher volume in the soma.

(12) Figure 2D, right: the term intoxication is improper for this experiment. Rather, it is the presence of the BoNT/A-mNG11 that is detected. I believe the authors should be particularly careful about the use of terms: intoxication means blockade of neurosecretion, SNAPR cleavage means activity etc.

While the reviewer is correct that it is the presence of BoNT/A-mNG11 that is detected, it remains that it is an active toxin, so the neurons are effectively intoxicated; as they are when we use the wild type toxin. We do not imply that we are measuring intoxication, but simply that the neurons are put into contact with a toxin.

(13) Line 196: Should we read TXNRD1 is required for BoNT/A LC translocation?TXNRD1 in the current model of translocation is located in the cytoplasm and is supposed to play a role in the cleavage of the disulfide bond linking LC to HC. In the model proposed by this study, LC is translocated through the Sec61 translocon. In this case, I would assume that the protein disulfide isomerase (PDI) in the endoplasmic reticulum would reduce the LC-HC disulfide bond. In that case, TXNRD1 would not be required anymore. Please discuss.

Why should we assume that a PDI is involved in the reduction of the LC-HC disulfide bond? In our previous studies on A-B toxins (PE and Ricin), different reduction systems seemed to be at play. There is no conceptual imperative to assume reduction in the ER because the Sec61 translocon is implicated. Reduction might occur on the cytosolic side by TXNRD1 or the effect of this reductase could be indirect.

(14) The legend of Figure 4 (in principle Figure 5?) is not matching with the panels and panel entries are missing (Figure 4F in particular).

Fixed

(15) Figure 6 panels E and H, please match colors with legend (grey and another color).

Not clear

(16) Please indicate BoNT/A construct concentrations in all Figure legends.

Done

(17) Line 416: isn't SV2 also involved in epilepsy?

Yes it is.

(18) Line 433: as above, shouldn't the disulfide bond linking LC to HC be cleaved by PDI in the ER in this model (as for other translocating bacterial toxins) rather than by thioredoxin reductases in the cytoplasm? Please discuss.

See above

(19) Identification of vATPase in the screen could be consistent with the endocytic vesicle acidification model of translocation.

Yes

(20) Did the authors add KCl in screening controls without toxins? This should be detailed in the Materials and Methods. Could there be a KCl effect on the cells? KCl exposure for 48 hours may be highly stressful for cells. The KCl exposure should last only several minutes for toxin entry.

We did not observe significant cell detah with the cell culture conditions used. Cell viability was controlled at multiple stages using nuclei number for instance

**Reviewer #3 (Recommendations For The Authors):**
Main comments:(1) In Figure 1B: could you devise a means to prevent proteosomal degradation of the tGFP cleaved part to assess whether this is formed?

We have also used a FRET assay after tintoxication and obtained similar results

(2) Line 152: Where it reads "was not surprising", maybe I missed something, but to me, this is indeed surprising. If the toxin is rapidly internalized and translocated (therefore, it is able to cleave SNAP25), the fact that tGFP requires 48 hours to be degraded seems surprising to me. Or does it mean that the toxin also slows down the degradation of the tGFP fragment? So, how can you differentiate between the effect being on cleavage of the fragment or in tGFP degradation?

The reviewer is correct, the “not” was a typo due to re-writting; the long delay between adding the toxin and observing cleavage was suprising indeed. Our interpretation is that it is trafficking that takes time, indeed, the split-GFP data kinetics indicates that the toxin takes about 48h to fill up the entire cytosol (Fig. 2D).

(3) Regarding the effect of Sec61G knockdown, is it possible that the observed effects are indirect and not due to the translocon being directly responsible for translocating the protein?

As discussed in the last part of the results,Sec61 knock-down results in block of intoxication, but does not prevent BonT from reaching the lumen of the ER (Figure 6G,H). Thus, Sec61 is “is instrumental to the translocation of BoNT/A LC into the neuronal cytosol at the soma.”

Minor comments:

(1) Fig. 3E: in the legend I think one of the NT3+ should be NT3-.

Yes, thanks for spotting it

(2) Would you consider adding Figure S4 as a main figure?

Thanks for the suggestion

(3) Please, check that all microscopy image panels have scale bars.

Done

(4) Figure 6B (bottom panes): why does it seem that there is a lot of mNeonGreen positive signal in regions that are not positive for HA? Shouldn't complementation keep HA in the complemented protein.

Our assumption i sthat there is an excess of receptor protein (HA tag) over reconstituted protein (GFP protein) given the relatively low concentration of toxin being internalized and translocatedRefs:(1) Pirazzini M, Azarnia Tehran D, Leka O, Zanetti G, Rossetto O, Montecucco C. On the translocation of botulinum and tetanus neurotoxins across the membrane of acidic intracellular compartments. Biochim Biophys Acta. 2016 Mar;1858(3):467–474. PMID: 26307528

(2) Pirazzini M, Rossetto O, Eleopra R, Montecucco C. Botulinum Neurotoxins: Biology, Pharmacology, and Toxicology. Pharmacol Rev. 2017 Apr;69(2):200–235. PMCID: PMC5394922

(3) Dong M, Masuyer G, Stenmark P. Botulinum and Tetanus Neurotoxins. Annu Rev Biochem. Annual Reviews; 2019 Jun 20;88(1):811–837.

(4) Rossetto O, Pirazzini M, Fabris F, Montecucco C. Botulinum Neurotoxins: Mechanism of Action. Handb Exp Pharmacol. 2021;263:35–47. PMCID: 6671090

(5) Williams JM, Tsai B. Intracellular trafficking of bacterial toxins. Curr Opin Cell Biol. 2016 Aug;41:51–56. PMCID: PMC4983527

(6) Mesquita FS, van der Goot FG, Sergeeva OA. Mammalian membrane trafficking as seen through the lens of bacterial toxins. Cell Microbiol. 2020 Apr;22(4):e13167. PMCID: PMC7154709

(7) Hoch DH, Romero-Mira M, Ehrlich BE, Finkelstein A, DasGupta BR, Simpson LL. Channels formed by botulinum, tetanus, and diphtheria toxins in planar lipid bilayers: relevance to translocation of proteins across membranes. Proc Natl Acad Sci U S A. 1985 Mar;82(6):1692–1696. PMCID: PMC397338

(8) Donovan JJ, Middlebrook JL. Ion-conducting channels produced by botulinum toxin in planar lipid membranes. Biochemistry. 1986 May 20;25(10):2872–2876. PMID: 2424493

(9) Fischer A, Montal M. Single molecule detection of intermediates during botulinum neurotoxin translocation across membranes. Proc Natl Acad Sci U S A. 2007 Jun 19;104(25):10447–10452. PMCID: PMC1965533

(10) Fischer A, Nakai Y, Eubanks LM, Clancy CM, Tepp WH, Pellett S, Dickerson TJ, Johnson EA, Janda KD, Montal M. Bimodal modulation of the botulinum neurotoxin protein-conducting channel. Proc Natl Acad Sci U S A. 2009 Feb 3;106(5):1330–1335. PMCID: PMC2635780

(11) Fischer A, Montal M. Crucial role of the disulfide bridge between botulinum neurotoxin light and heavy chains in protease translocation across membranes. J Biol Chem. 2007Oct 5;282(40):29604–29611. PMID: 17666397

(12) Koriazova LK, Montal M. Translocation of botulinum neurotoxin light chain protease through the heavy chain channel. Nature structural biology. 2003. p. 13–18. PMID: 12459720

(13) Moreau D, Kumar P, Wang SC, Chaumet A, Chew SY, Chevalley H, Bard F.Genome-wide RNAi screens identify genes required for Ricin and PE intoxications. Dev Cell. 2011 Aug 16;21(2):231–244. PMID: 21782526

(14) Bassik MC, Kampmann M, Lebbink RJ, Wang S, Hein MY, Poser I, Weibezahn J, Horlbeck MA, Chen S, Mann M, Hyman AA, Leproust EM, McManus MT, Weissman JS. A systematic mammalian genetic interaction map reveals pathways underlying ricin susceptibility. Cell. 2013 Feb 14;152(4):909–922. PMCID: PMC3652613

(15) Tian S, Muneeruddin K, Choi MY, Tao L, Bhuiyan RH, Ohmi Y, Furukawa K, Furukawa K, Boland S, Shaffer SA, Adam RM, Dong M. Genome-wide CRISPR screens for Shiga toxins and ricin reveal Golgi proteins critical for glycosylation. PLOS Biol. 2018 Nov;16(11):e2006951. PMCID: PMC6258472